# Inter-cell type interactions that control JNK signaling in the *Drosophila* intestine

Peng Zhang [1] ✉, Stephen M. Pronovost[1], Marco Marchetti[1], Chenge Zhang[1], Xiaoyu Kang[1], Tahmineh Kandelouei[1], Christopher Li[1,2] & Bruce A. Edgar [1] ✉

JNK signaling is a critical regulator of inflammation and regeneration, but how it is controlled in specific tissue contexts remains unclear. Here we show that, in the *Drosophila* intestine, the TNF-type ligand, Eiger (Egr), is expressed exclusively by intestinal stem cells (ISCs) and enteroblasts (EBs), where it is induced by stress and during aging. Egr preferentially activates JNK signaling in a paracrine fashion in differentiated enterocytes (ECs) via its receptor, Grindelwald (Grnd). *N*-glycosylation genes (*Alg3*, *Alg9*) restrain this activation, and stress-induced downregulation of *Alg3* and *Alg9* correlates with JNK activation, suggesting a regulatory switch. JNK activity in ECs induces expression of the intermembrane protease Rhomboid (Rho), driving secretion of EGFR ligands Keren (Krn) and Spitz (Spi), which in turn activate EGFR signaling in progenitor cells (ISCs and EBs) to stimulate their growth and division, as well as to produce more Egr. This study uncovers an *N*-glycosylation-controlled, paracrine JNK-EGFR-JNK feedforward loop that sustains ISC proliferation during stress-induced gut regeneration.

The tumor necrosis phenomenon has been appreciated for hundreds if not thousands of years, as evidenced in writings from the ancient Egyptians and medieval scholars, who found that infectious stimuli could induce spontaneous tumor regression[1,2]. In the early 1900s a surgeon, William Coley, developed bacterial vaccines that stimulated the immune system to generate cytotoxic factors for anti-sarcoma treatment[1], and in the 1970s the first tumor necrosis factor (TNF-α) was isolated from the serum of endotoxin-treated mice[3]. The human TNF superfamily comprises at least 19 ligands and 29 receptors (TNFR) that regulate numerous pathophysiological processes, including auto-immune diseases, neurological diseases, and cancer[4]. Given the existence of these numerous ligands, receptors, and diverse intercellular interactions, the signaling pathways utilized by TNF/TNFR in mammals are complex and incompletely understood. At least five intracellular signaling cascades can be activated by TNF-α: apoptosis, extracellular signal-regulated kinase (ERK), p38 mitogen-activated protein kinase, NF-κB, and Jun N-terminal kinase (JNK)[4].

In *Drosophila*, the TNF/TNFR signaling system is simpler, comprising a single TNF-type ligand, Eiger (Egr)[5,6], and two receptors,

Wengen (Wgn)[7] and Grindelwald (Grnd)[8]. Egr is a type II transmembrane protein, which can be cleaved by the TNF-α converting enzyme, which releases it from the cell surface as a soluble cytokine that can function in autocrine, paracrine, and endocrine manners[9,10]. A recent study reported that Grnd has a much higher affinity than Wgn for Egr and that Grnd and Wgn have distinct subcellular localizations, suggesting that these two receptors respond to different thresholds of Egr to promote non-redundant cellular functions[11]. Another recent study showed that the N-linked glycosylation of Grnd decreases the Egr/Grnd-binding affinity, and thereby attenuates downstream signal transduction[12]. Like many human TNFs, *Drosophila* Egr is a potent inducer of apoptosis, and its pro-apoptotic function completely depends on its ability to activate the JNK pathway[5,10].

JNK signaling can be activated by environmental stresses (e.g., UV radiation, oxidative damage) and pro-inflammatory factors (e.g., TNFs), and plays critical roles in many cellular events including apoptosis, cell proliferation, and transformation. In *Drosophila*, JNK signaling can be initially transduced by any of four Jun-kinase kinase kinases (JNKKKs): TGFβ-associated kinase 1 (Tak1), apoptotic signal-regulating

[1]Huntsman Cancer Institute and Department of Oncological Sciences, University of Utah, Salt Lake City, UT 84112, USA. [2]Harvard University, Cambridge, MA 02138, USA. ✉e-mail: peng.zhang@hci.utah.edu; bruce.edgar@hci.utah.edu

kinase 1 (Ask1), Slipper (Slpr), or Wallenda (Wnd), that may sense different types of cellular stress and damage (reviewed in ref. 13). Among these, Tak1, Slpr, and Wnd, but not Ask1, can sense upstream signaling from Egr to Grnd or Wgn[13–16]. The JNKKKs phosphorylate and activate two JNK kinases (JNKKs), MAPK kinase 4 and Hemipterous (Hep), which subsequently phosphorylate the sole JNK in *Drosophila*, Basket (Bsk)[13]. Phosphorylated Bsk activates several transcription factors including AP-1, a heterodimer comprised of Jun-related antigen (Jra, also known as c-Jun) and Kayak (Kay, also known as Fos), to trigger target genes' expression[13]. One of AP-1's downstream targets, Puckered (Puc) encodes a JNK-phosphatase that dephosphorylates Bsk and acts as a negative feedback inhibitor[17]. Therefore, a *puc-lacZ* reporter is widely used as an indicator of JNK signaling activity in *Drosophila*[18–20]. Numerous studies have suggested that JNK signaling in *Drosophila* can be both pro-apoptotic and pro-proliferative and that the balance of these outputs depends on cellular context[13,21]. Typically, autocrine JNK signaling promotes cell death, whereas paracrine JNK signaling from TNF/TNFR triggers cell proliferation[21].

In the *Drosophila* midgut, the JNK pathway plays a pivotal role in regulating gut regeneration during aging or in response to acute stress, such as enteric infections. During aging of the gut, JNK activity typically rises in both mature enterocytes (ECs) and progenitor cells (intestinal stem cells (ISCs) and enteroblasts (EBs)). Several studies have suggested that changes in the composition and increased overall load of intestinal microbiota in aged guts (referred to as "dysbiosis") activate dual oxidase signaling in ECs, leading to the production of reactive oxygen species (ROS) that activate JNK signaling[18,19,22,23]. Consistent with this idea, feeding flies paraquat (an ROS-producing agent) or $H_2O_2$ activates JNK signaling in both ECs and progenitors[18,24,25]. Yet, precisely how elevated ROS triggers JNK activation remains unclear. Activation of JNK in progenitors induces the expression of stress-responsive genes to protect the gut from oxidative damage, stimulates ISC proliferation, and promotes ISC symmetric divisions[18,19,26]. JNK in ECs has different functions, such as eliminating damaged ECs to potentiate gut regeneration[20,27] and stimulating the expression of leptin/IL6-like cytokines (Unpaired 2 (Upd2), Unpaired 3 (Upd3)) that promote ISC proliferation through JAK/STAT and EGFR/MAPK signaling[19,20,28]. Importantly, while JNK can be activated throughout the ISC–EB–EC lineage during aging, its activation predominantly occurs in ECs[18]. Following acute gut damage (e.g., enteric infection by *Erwinia carotovora carotovora strain 15 (Ecc15*)), although JNK activation can be observed in progenitors, its activation in ECs is stronger (see "Results" section). A similar phenotype was observed in other independent studies, which showed that JNK activation is exclusively induced in ECs by enteric *Pseudomonas entomophila* (*P.e.*) or *Pseudomonas aeruginosa* infections[20,27]. Why ECs have a relatively lower threshold of JNK activation, and also whether Egr and Grnd/Wgn are required for JNK activation in the gut have remained unanswered questions.

In this study, we identify previously unknown cell type-specific patterns of JNK signaling between cells in the ISC–EB–EC lineage. We report that the JNK ligand Egr is exclusively expressed by progenitor cells, while JNK signaling is preferentially activated in ECs. JNK activity is restrained by ALG3 and ALG9, enzymes that catalyze N-linked glycosylation of Grnd to inhibit Egr/Grnd interaction. During stress, *Alg3* and *Alg9* expression is downregulated, and we propose that this facilitates Egr/Grnd interaction and JNK activation in the entire ISC–EB–EC lineage. The Grnd-dependent activation of JNK in ECs upregulates Rhomboid (Rho), which in turn activates the EGFR ligands, Keren (Krn) and Spitz (Spi), through proteolytic cleavage to allow their secretion. Finally, EC-derived Krn and Spi trigger EGFR signaling in progenitor cells to produce more Egr. Altogether, our work uncovers a previously unknown Egr/Grnd/JNK-Rho-Krn/Spi-EGFR-Egr feedforward signaling interaction between ECs and progenitors that sustains the production of Egr and JNK activity during gut epithelial regeneration, but which could also contribute to chronic inflammation or tumor progression.

## Results

### Eiger is induced in progenitor cells by aging and acute gut damage

To investigate the function of Egr, we first examined its expression in adult fly midguts using an endogenous GFP-tagged Egr (a MiMIC line[29]). In adults 1 day after eclosion, the expression of Egr-GFP was barely detected (Fig. 1a, a'). However, at 10 and 20 days after eclosion, a progressive increase in Egr-GFP was observed in gut progenitor cells labeled by *esg-lacZ* (Fig. 1b–d'). Further, we found that Egr could be markedly induced in progenitors in 1-day-old flies by enteric infection with *P.e.* (Fig. 1e, e'). Because *P.e.* infection causes rapid cell turnover in the gut, we also observed faint expression of Egr in newborn ECs (nECs) in this condition (Fig.1f, f', Pdm1+ cells), suggesting that low levels of Egr-GFP were inherited from EBs. These data indicate that Egr expression is specific to progenitor cells, and is induced by aging and gut damage, consistent with previous observations that JNK activity is upregulated by aging and gut damage[18–20].

### JNK activation is restricted to differentiated enterocytes and enteroendocrine cells

Given the inducible, progenitor-specific expression of Egr, we investigated the activation pattern of JNK signaling in the gut using the *puc-lacZ[E69]* reporter, a widely used downstream target of JNK[18–20]. Under normal conditions at 8 days after eclosion, *puc-lacZ[E69]* showed weak expression in large cells (Fig. 2a, white arrowheads) that did not express *egr-GFP* (Fig. 2a, yellow arrowheads), namely ECs. In flies older than 5 days post eclosion, *puc-lacZ[E69]* was also observed in enteroendocrine (EE) cells identified by the EE marker, Prospero (Pros) (Fig. 2b, white arrows). We also examined another JNK pathway reporter, *TRE-DsRed*[30], and found that this was also specifically expressed in ECs (Supplementary Fig. 1, white arrowheads). These data demonstrate that JNK is exclusively activated in ECs and EEs and remains inactive in progenitor cells under normal conditions. Since previous studies reported that JNK could be activated in progenitor cells under stress conditions[18,19], we examined the expression of *puc-lacZ[E69]* after enteric infection with *Ecc15*. Although infection could induce low levels of *puc-lacZ[E69]* expression in progenitors (Fig. 2c, right panels, yellow arrowheads), ECs showed much stronger *puc-lacZ[E69]* induction in this condition (Fig. 2c, right panels, white arrowheads). These data indicate that JNK activation in progenitors is significantly restricted in both normal and stress conditions. To learn about the mechanisms of cell type-specific JNK activation, we further investigated the functions of Egr and its receptors, Grnd and Wgn.

### Progenitor cell-produced Eiger signals to Grindelwald in differentiated enterocytes

Previous studies have shown that hemocytes[31] and fat body cells[9] can also produce Egr and that those two cell types can actively communicate with the gut[32,33]. Therefore, we next evaluated whether gut progenitor cells are the major source of Egr for JNK activation in the gut. RNAi-mediated knockdown of *egr* either in hemocytes (*Hml[ts]*-driven, Supplementary Fig. 2a) or in fat body (*Lpp[ts]*-driven, Supplementary Fig. 2b) had no repressive effect on damage-induced ISC hyperproliferation. This suggested that gut-specific Egr may play a more important role in JNK activation. Although Egr is produced by gut progenitor cells, overexpression of *egr* in these cells did not promote ISC proliferation (Fig. 3a, n)[34]. Also, in contrast to other JNK pathway components (e.g., *grnd*, *Hep[Act]*, and *puc[RNAi]*), overexpression of *egr* in ECs (*Myo1A[ts]*-driven) also failed to promote ISC mitoses (Fig. 3d). Hence Egr expression is not sufficient to promote ISC proliferation. However, RNAi-mediated knockdown of *egr* in progenitors did significantly repress the ISC proliferation caused by gut damage (Fig. 3b), and overexpressed *egr* increased the mitogenic effect of gut damage (Fig. 3c). These data indicate that Egr is more effective under stress conditions, raising the possibility that stress may somehow enhance

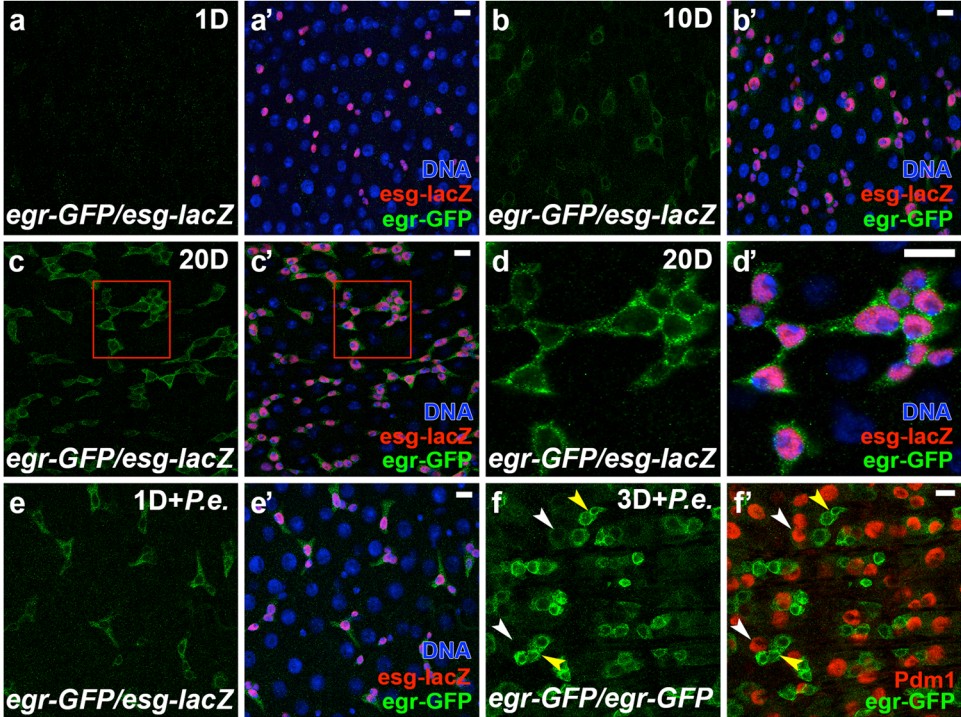

**Fig. 1 | Eiger is induced in progenitor cells by aging and acute gut damage.** Flies were raised at 25 °C. The *egr-GFP/esg-lacZ* adult female flies of ages 1 day (**a**, **a′**), 10 days (**b**, **b′**), and 20 days (**c**, **c′**) were dissected. Midguts were stained with anti-GFP/-lacZ antibodies. **d**, **d′** Enlarged views of the red-boxed regions in (**c**, **c′**). **e**, **e′** One-day-old *egr-GFP/esg-lacZ* adult females were orally infected with *Pseudomonas entomophila* (*P.e.*) for 16 h and subsequently dissected. Midguts were stained with anti-GFP/-lacZ antibodies. *Esg-lacZ* was used to label gut progenitors (ISCs and EBs). Nuclei were labeled in blue. **f**, **f′** Three-day-old *egr-GFP/egr-GFP* adult females were infected with *P.e.* for 18 h and then dissected. Midguts were stained with anti-GFP/-Pdm1 (EC marker) antibodies. Yellow arrowheads indicate progenitors and white arrowheads indicate ECs. Images in (**a–f′**) are representative of three independent experiments. Scale bars: 10 μm.

the signaling capability of Egr. In addition, although the JNK reporter *puc-lacZ^{E69}* showed activation in EEs (Fig. 2b), overexpressing *egr* or its receptor *grnd* in EEs did not have any obvious cell non-autonomous pro-mitotic effects on ISCs (Fig. 3e).

Our investigation next aimed to determine which receptor (Grnd or Wgn) responds to Egr signaling in the gut. Analyses of RNA-sequence (RNA-seq) datasets from our laboratory revealed higher *wgn* mRNA levels compared to *grnd* in the adult midgut under normal conditions (Fig. 3f), but after *P.e.* infection *grnd* mRNA significantly increased, while *wgn* mRNA decreased (Fig. 3f). To test the requirements for these receptors, we examined the loss-of-function effects of *grnd* and *wgn* in the ISC–EB–EC lineage during stress conditions, when JNK is activated and sufficient to trigger ISC proliferation. RNAi-mediated knockdown of *grnd* in progenitors (*esg^{ts}*-driven), ECs (*mex^{ts}*-driven), or the EB–EC lineage (*Su(H)GBE^{ts}-FlipOut*-driven), all significantly decreased stress-induced ISC proliferation, whereas *wgn* depletion in any of these cell types did not show suppressive effects (Fig. 3g, h, Supplementary Fig. 3a–c). Using more specific Gal4 drivers (*Dl^{ts}* for ISCs and *Su(H)GBE^{ts}* for EBs) to further pinpoint the locus of Grnd function, we found that depletion of *grnd* in ISCs had no effect (Fig. 3i), whereas depletion in EBs significantly inhibited stress-induced ISC mitoses (Fig. 3j). Knockdown of *wgn*, however, had no inhibitory effect with either driver (Fig. 3i, j).

Next, we examined the gain-of-function effects of *grnd* and *wgn* in the gut. Overexpressing *grnd* using the *esg^{ts}* driver significantly increased ISC proliferation (Fig. 3k). However, overexpression of *wgn* using the same driver had no obvious pro-mitotic phenotype (Fig. 3l). In fact, ectopic *wgn* markedly decreased damage-induced ISC hyper-proliferation (Fig. 3l). These data emphasize that Grnd, rather than Wgn, serves as the relevant receptor for triggering ISC divisions in the gut stem cell lineage, and suggest that the levels of *grnd* in EBs and ECs exert profound effects on JNK-dependent stem cell proliferation.

We further detailed the gain-of-function effects of *grnd* in the ISC–EB–EC lineage using more specific drivers. Consistent with *grnd*'s loss-of-function effects, overexpression of *grnd* in ISCs was not pro-mitotic (Fig. 3m), whereas overexpressing *grnd* in EBs (Fig. 3n), ECs (Fig. 3d), or the EB–EC lineage (Supplementary Fig. 3d, e) elicited strong pro-proliferative effects. These gain-of-function effects align with our observations in the loss-of-function assays and further affirm Grnd's crucial role in EBs and ECs for JNK activation. Finally, we found that the gain-of-function phenotype of Grnd is Egr-dependent (Fig. 3k, n). Taken together, these data indicate that signaling between Egr, produced by progenitor cells, to Grnd in EBs and ECs, is important for ISC proliferation.

To extend this analysis, we tested Grnd's ability to activate *puc-lacZ^{E69}*. Tellingly, overexpression of *grnd* in progenitors did not upregulate *puc-lacZ^{E69}* in these cells (Fig. 3o, yellow arrowheads) but preferentially and strongly upregulated *puc-lacZ^{E69}* in ECs (Fig. 3o, white arrowheads). This suggests that Grnd's ability to activate JNK is confined to ECs and its activity is repressed in ISCs and EBs. However, a question arises regarding why overexpression of *grnd* in EBs, driven either by *esg^{ts}* (Fig. 3k) or *Su(H)GBE^{ts}* (Fig. 3n), could induce significant ISC mitoses. One plausible explanation is that, due to the transient nature of EBs, which rapidly differentiate into ECs during gut epithelial regeneration, overexpression of *grnd* in EBs results in nECs inheriting the EB-overexpressed Grnd. Consequently, the increased JNK activity caused by ectopic Grnd in these nECs may trigger additional mitogenic signaling to stimulate ISC proliferation in a cell non-autonomous manner.

**Hyperactivation of JNK signaling causes apoptotic stem cell loss**
We then asked why progenitor cells are refractory to JNK activation. It's well known that overactivation of JNK is strongly pro-apoptotic in many tissues. However, in the fly gut, activation of JNK promotes stem

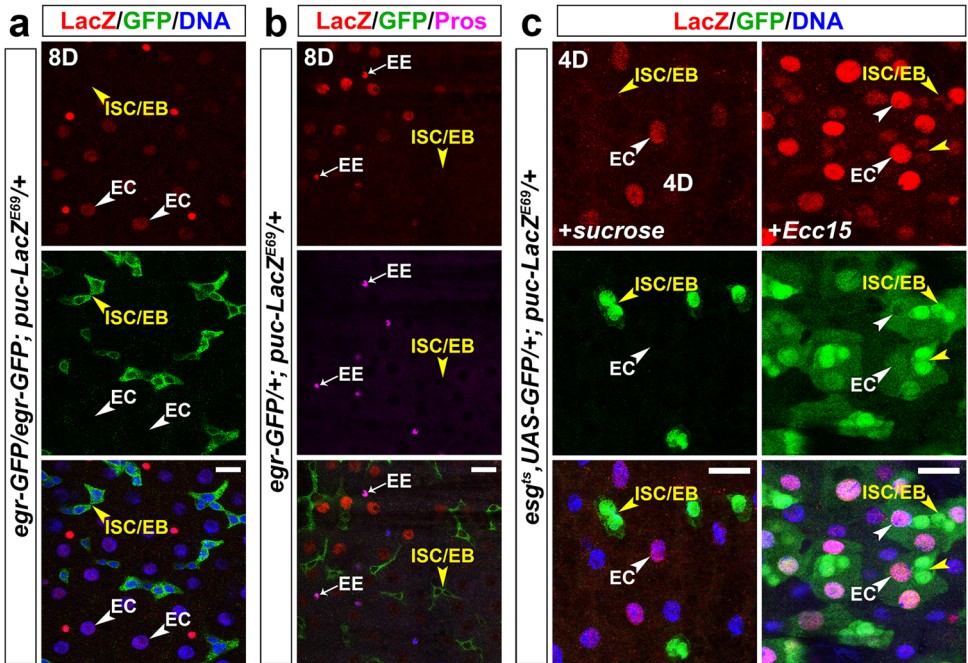

**Fig. 2 | JNK activation is restricted to differentiated ECs and EEs.** Flies were raised at 25 °C. *Puc-lacZ^E69* was used to indicate the activation of JNK signaling. Eight-day-old *egr-GFP/egr-GFP; puc-lacZ^E69/+* (**a**) or *egr-GFP/+; puc-lacZ^E69/+* (**b**) adult female flies were dissected. Midguts were stained with anti-GFP/-lacZ/-Pros anti-bodies as indicated in panels. Yellow arrowheads indicate progenitors (ISCs/EBs), white arrowheads in (**a**) indicate ECs, and white arrows in (**b**) indicate EEs. **c** Flies were raised at 18 °C. One-day-old *esg^ts; UAS-GFP/+; puc-lacZ^E69/+* adult female flies

were shifted to 29 °C for 4 days and then fed with 5% sucrose (control, left panels) or *Ecc15* (right panels) for 18 h before dissection. Midguts were stained with anti-GFP/-lacZ antibodies. Yellow arrowheads indicate ISCs/EBs and white arrowheads indicate ECs. Nuclei were labeled in blue. Images in (**a**, **b**) are representative of three independent experiments. Images in (**c**) are representative of two independent experiments. Scale bars: 15 μm.

cell mitoses[18–20,27]. To investigate this, we tested the pro-mitotic effect of JNK activation by overexpressing a constitutively active form of the JNKK *Hep, Hep^Act*. Consistent with previous observations, over-expressing *Hep^Act* using *Dl^ts*, a weak ISC-specific driver, induced high levels of ISC proliferation (Fig. 4a). When we used the stronger *esg^ts* driver, which expresses in both ISCs and EBs, the pro-mitotic effect of *Hep^Act* appeared to be absent. However, co-expressing the caspase inhibitory protein *P35* with *Hep^Act* could somewhat restore *Hep*'s pro-mitotic effect at an early timepoint (1 day; Fig. 4b). But over time (e.g., 3-day overexpression), *P35* lost this rescue capability (Fig. 4b). These data suggest that high levels of JNK activity negate ISCs' mitotic cap-ability, and that this effect is dose-dependent. Further tests using the *esg^ts* driver showed that *Hep^Act*-expressing midguts had lost most of their pre-existing GFP⁻ ECs, which were replaced by newborn GFP⁺ ECs (Fig. 4c, d), and that progenitor cell numbers were strongly reduced over time (Fig. 4e). This suggests that *Hep^Act* expression triggered rapid cell turnover before causing progenitor loss. To validate this, we uti-lized the Gal4 Technique for Real-time And Clonal Expression (G-TRACE) system[35] to track the cell fate of *Hep^Act*-expressing ISCs.

The G-TRACE cell lineage-tracing system (*UAS-RFP, UAS-Flp, Ubi^p63E > FRT-STOP-FRT > nEGFP*) was activated by temperature shift using *esg^ts*. In this method, *UAS-RFP* and *UAS-Flp* are induced in pro-genitors by *esg-Gal4*. FLP recombinase excises the *FRT-STOP-FRT* cas-sette from *Ubi^p63E > FRT-STOP-FRT > nEGFP*, converting it to a ubiquitously expressed GFP marker, *Ubi^p63E-nEGFP*. As a result, all progenitor cells are marked strongly with RFP and weakly with GFP (Fig. 4f, yellow arrowheads in upper panels) while their progeny cells are marked with GFP only. Under normal conditions, due to active gut regeneration, nECs inherited some RFP signal from their ISC/EB pre-cursors, thereby exhibiting a mild-RFP/strong-GFP double positive pattern (Fig. 4f, white arrowheads in upper panels). In contrast, older pre-existing ECs (oECs) exhibited an RFP/GFP double negative pattern

(Fig. 4f, white arrows in upper panels). Remarkably, in *Hep^Act* over-expression guts RFP⁺ progenitor cells disappeared and the number of nECs significantly decreased (Fig. 4f, white arrowheads in lower panels). These data confirm that hyperactivation of JNK in progenitors causes stem cell loss and impairs gut regeneration. Indeed, high levels of the apoptosis marker, DCP-1, were evident in the *Hep^Act*-expressing progenitors (Fig. 4g, yellow arrowheads in right panels). These experiments demonstrate that low levels of JNK activity in ISCs pro-mote ISC proliferation whereas high levels trigger apoptosis (Fig. 4h). This biphasic function provides a rationale for why JNK signaling is restricted to differentiated cells, namely because too much JNK acti-vation in ISCs is cytotoxic.

Additionally, we explored whether JNK activation in ECs can also trigger apoptosis and whether it has dose-dependent functions in these cells too. In contrast to progenitor cells, ECs exhibited hyper-sensitivity to JNK activation. A brief period (1 day) of *Hep^Act* expression in ECs induced robust EC apoptosis, as indicated by DCP-1 staining (Supplementary Fig. 4a). However, over time (e.g., 3-day over-expression), there was no significant difference in the degree of EC apoptosis (Supplementary Fig. 4a, b), suggesting that JNK triggers EC apoptosis in a dose-independent manner.

## JNK signaling in ISCs is restricted by *N*-glycosylation

Continuing our exploration, we aimed to decipher the mechanisms responsible for suppressing JNK activation in progenitor cells. A recent study highlighted the role of the Gish/CK1r kinase in restraining JNK activation by phosphorylating and destabilizing Rho1, a JNK activator[36]. Gish is downregulated in midgut progenitors in older flies, leading to increased JNK activity during aging[36]. However, knockdown of *Gish* in progenitors for 3 days did not significantly increase ISC proliferation[36], suggesting that other factors are involved. Another recent study repor-ted that the *N*-glycosylation pathway components ALG3 and ALG9

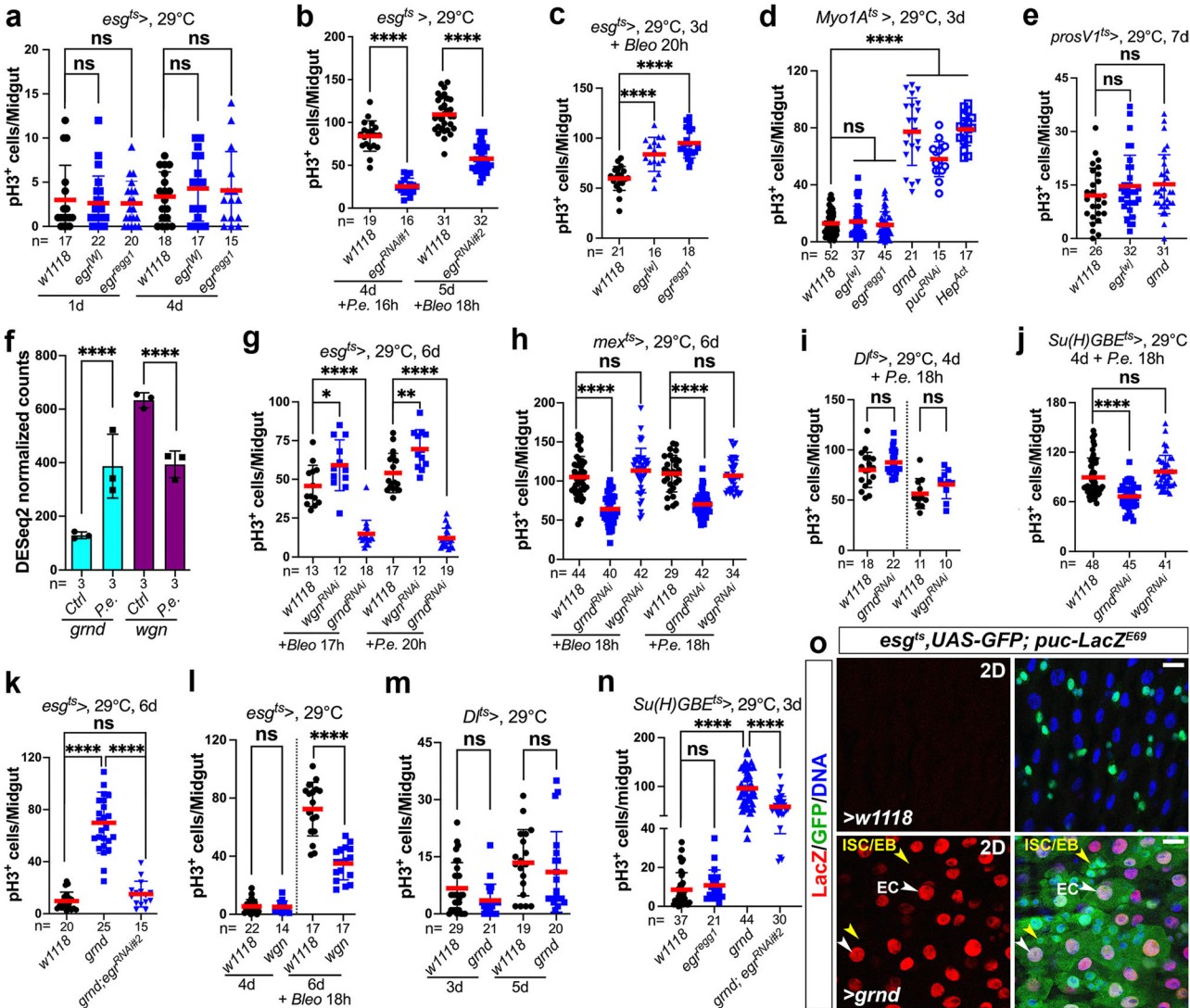

**Fig. 3 | Progenitor cell-produced Egr signals to Grnd in differentiated ECs.**
Genetic manipulations targeting components of the JNK pathway in different cell types of adult fly midguts were conducted with the following drivers: *Dl^ts* for ISCs (**i**, **m**), *esg^ts* for progenitors (ISCs and EBs) (**a**–**c**, **g**, **k**, **l**, and **o**), *Su(H)GBE^ts* for EBs (**j**, **n**), *Myo1A^ts* (**d**) or *mex^ts* for ECs (**h**), and *prosV1^ts* for EEs (**e**). Flies were raised at 18 °C and 2–3-day-old adult females were shifted from 18–29 °C for various durations (as indicated in panels) before treatments (*P.e.* or 250 μM bleomycin) and subsequent dissections. Midguts were stained with anti-pH3 antibodies, and ISC mitoses were quantified by counting pH3^+ cells. Quantifications of data shown in (**a**–**e** and **g**–**n**) represent the mean ± SD (two-tailed unpaired *t*-test, $^{ns}P > 0.05$, $^*P = 0.0356$ (**g**), $^{**}P = 0.0032$ (**g**), $^{****}P < 0.0001$). *N* values in individual panels indicate the number of midguts examined. **a** *UAS-egr^{[w]}* represents a standard *UAS* line

with the *egr* cDNA is cloned into the *pUAST* vector[5], while *UAS-egr^{regg1}* is a GS transposon insertion line (GS9830) that contains UAS enhancers inserted into the promoter region of *egr*[5]. **f** The expression levels of *grnd* and *wgn* were examined using available RNA-seq datasets from our laboratory. DESeq2 was used for count normalization (plotted) and differential expression analysis (*n* = 3 replicates, error bars indicate mean ± SD, **** adjusted *p* value < 0.0001) of *grnd* and *wgn* in the adult fly midgut with/without *P.e.* infection. **o** Overexpression of *grnd* in progenitors was driven by *esg^ts* at 29 °C for 2 days. Midguts were stained with anti-GFP/-lacZ antibodies, with nuclei labeled in blue. *Puc-lacZ^{E69}* was used to indicate the activation of JNK signaling. Yellow arrowheads indicate ISCs/EBs and white arrowheads indicate ECs. Images in (**o**) are representative of two independent experiments. Scale bars: 15 μm. Source data are provided as a Source Data file.

repress JNK activity in fly imaginal discs[12]. ALG3 and ALG9 act as alpha-1,3-mannosyltransferases that modify proteins during the process of *N*-glycosylation. Among the five mannosyltransferase-encoding genes in *Drosophila* (*Alg2/CG1291*, *Alg3/CG4084*, *Alg9/CG11851*, *Alg11/CG11306*, and *Alg12/CG8412*), our screening showed that RNAi-mediated knockdown of either *Alg3* or *Alg9* in progenitors increased ISC mitoses (Fig. 5a), implicating these two genes as potential regulators of JNK activity. To discern the cell types in which ALG3 and ALG9 exert their anti-mitotic functions, we employed specific Gal4 drivers – *Dl^ts* for ISC knockdown, *Su(H)GBE^ts* for EB knockdown, and *Myo1A^ts* for EC knockdown. Remarkably, depletion of *Alg3* or *Alg9* in each cell type resulted in ISC over-proliferation (Fig. 5c–e), indicating that ALG3 and ALG9 repress ISC proliferation, both cell autonomously and non-cell autonomously.

To determine whether ALG3 and/or ALG9 repress ISC proliferation by restricting JNK activity, we examined the induction of *puc-lacZ^{E69}* after *Alg3*- or *Alg9*-depletion. Knockdown of either gene using *esg^ts* markedly induced *puc-lacZ^{E69}* expression in progenitors (Fig. 5b) as well as in newborn (GFP^+) ECs, confirming that ALG3 and 9 are important suppressors of JNK activity. We next investigated whether the ISC hyperproliferation caused by *Alg3*- or *Alg9*-depletion is JNK dependent. Expression of *egr^{RNAi}* significantly blocked ISC mitoses caused by *Alg3^{RNAi}* (Fig. 5c, d), and expression of *Bsk^{KS3R}* (a dominant-negative form of Bsk/JNK) significantly suppressed ISC mitoses caused by *Alg9^{RNAi}* (Fig. 5c). Interestingly, inactivating JNK signaling in ECs with *Bsk^{KS3R}* was insufficient to suppress *Alg9^{RNAi}*'s pro-mitotic effects (Fig. 5e), suggesting that ALG9 and/or ALG3 may have multiple

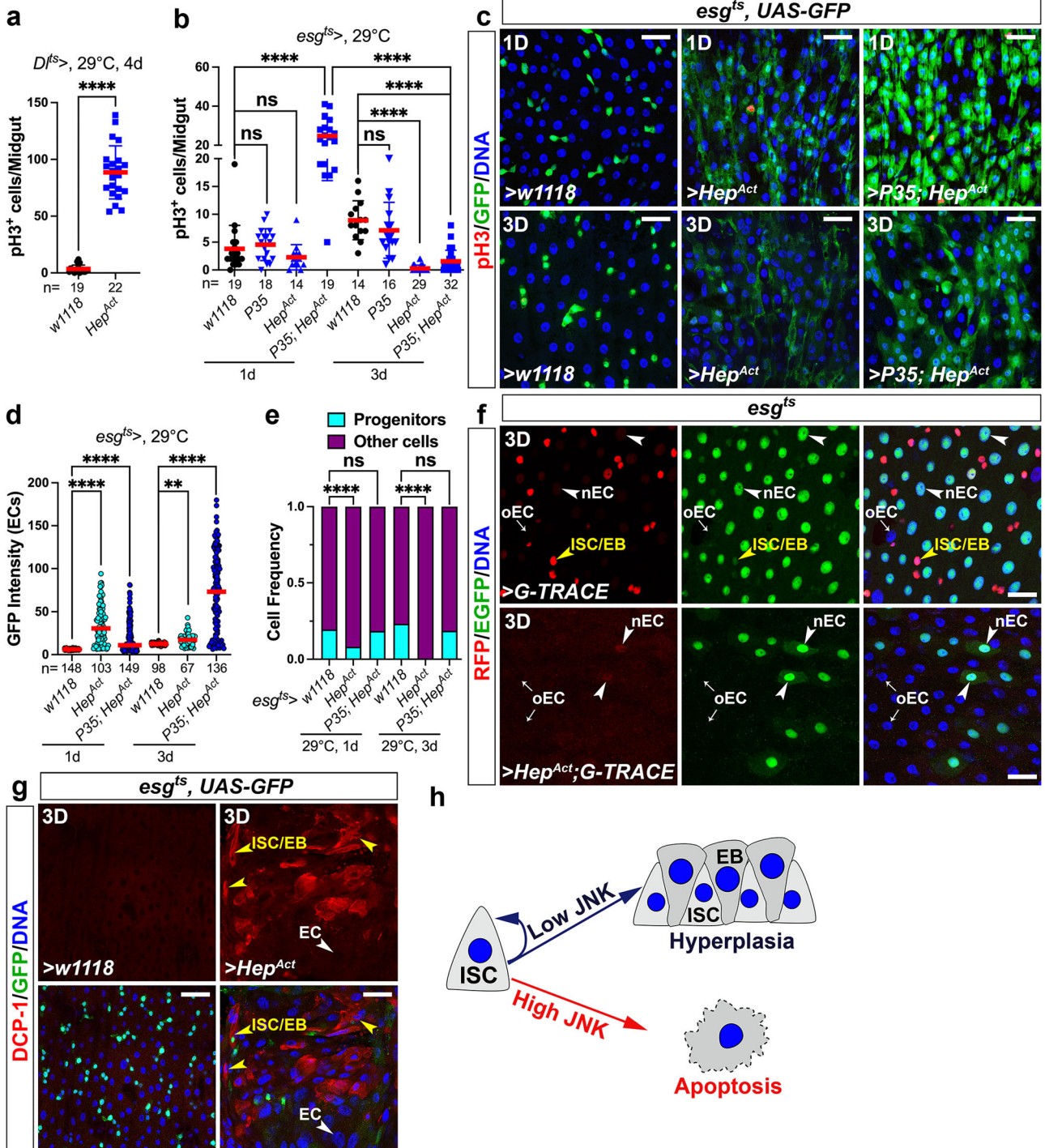

**Fig. 4 | Hyperactivation of JNK signaling causes apoptotic stem cell loss.**
**a**, **b** Adult female midguts were stained with anti-pH3 antibodies, and ISC mitoses were quantified by counting pH3+ cells. Quantification data represent the mean ± SD (two-tailed unpaired *t*-test, ns*p* > 0.05, ****p < 0.0001), with each dot representing one sample. **c**, **g** *w1118* (control), *UAS-Hep^Act^*, or *UAS-P35; UAS-Hep^Act^* was overexpressed in progenitors driven by *esg^ts^*. Flies were raised at 18 °C and then shifted to 29 °C for 1 or 3 days before dissection. Midguts were stained with anti-GFP/-pH3 (**c**) or anti-GFP/-DCP-1 (**g**) antibodies. **d** The GFP intensities in newborn ECs (GFP+) of each genotype were quantified and compared (two-sided Dunn's test, **p = 0.0023, ****p < 0.0001, *N* values indicate the number of ECs examined). **e** The frequencies of GFP+ progenitors were quantified and compared (Fisher's exact test with

Benjamini–Hochberg correction for multiple testing, ns*p* > 0.05, ****p < 0.0001).
**f** *G-TRACE* or *Hep^Act^* + *G-TRACE* was overexpressed using *esg^ts^* at 29 °C for 3 days. Midguts were stained with anti-GFP/-DsRed antibodies. Progenitors are marked with yellow arrowheads, newborn ECs (nECs) with white arrowheads, and old ECs (oECs) with white arrows. **g** Progenitors are denoted by yellow arrowheads and ECs by white arrowheads. Scale bars: 30 μm in (**c**, **g**) and 20 μm in (**f**). Nuclei were labeled in blue. Images in (**c**) are representative of two independent experiments. Images in (**f**, **g**) are representative of three independent experiments. **h** The data from (**a**–**g**) indicated a model that long-time hyperactivation of JNK signaling in progenitors causes apoptosis, whereas transient overactivation of JNK promotes ISC proliferation. Source data are provided as a Source Data file.

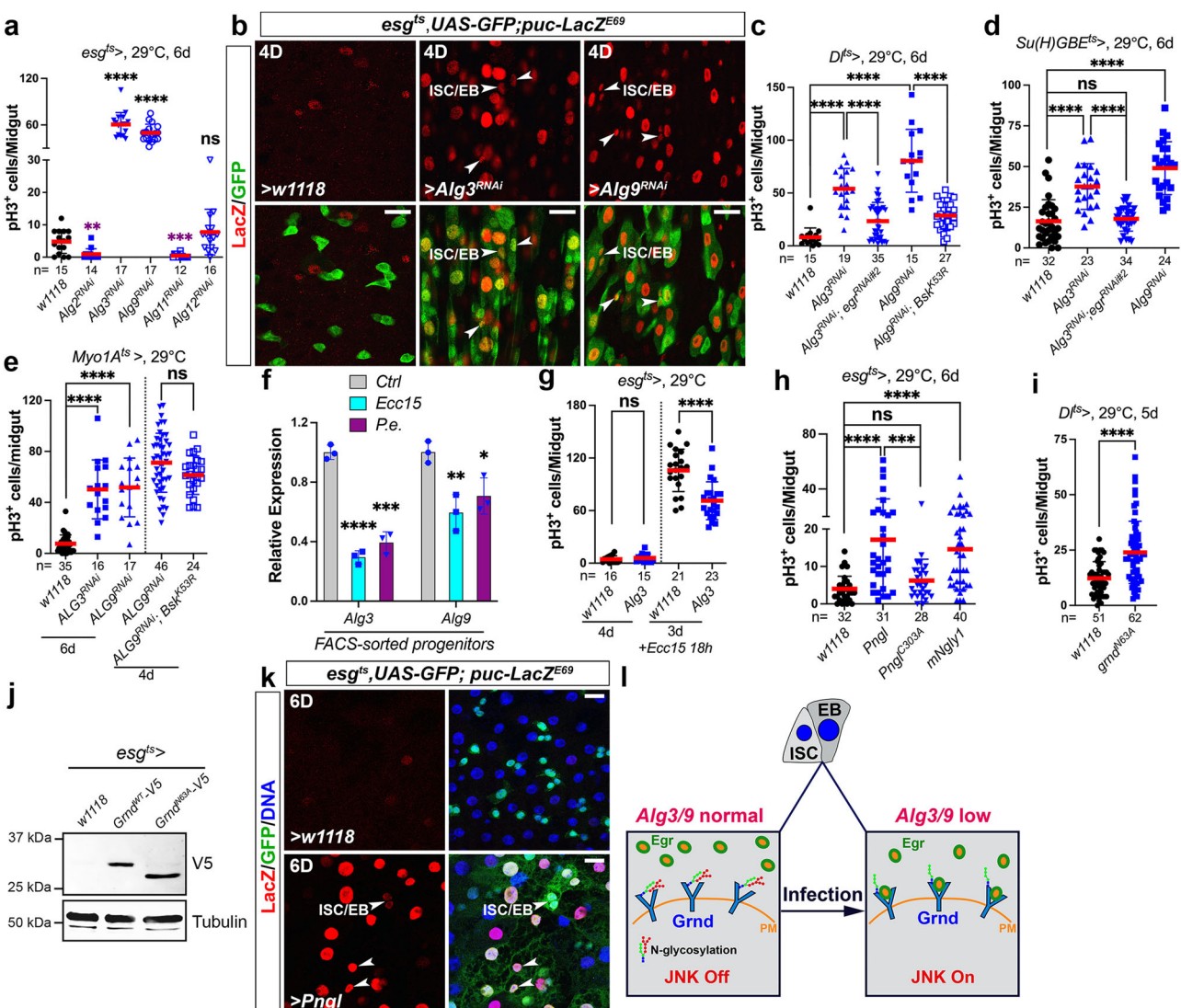

**Fig. 5 | The activation of JNK signaling in ISCs is restricted by *N*-glycosylation.**
**a**, **c**–**e**, **g**–**i** Different genetic manipulations are indicated in individual panels. Midguts were stained with anti-pH3 antibodies, and ISC mitoses were quantified by counting pH3⁺ cells. Quantification data represent the mean ± SD (two-tailed unpaired *t*-test, ⁿˢ*P* > 0.05, ***P* = 0.0012 (**a**), ****P* = 0.0005 (**a**), ****P* = 0.001 (**h**), *****P* < 0.0001). *N* values in individual panels indicate the number of midguts examined. **b**, **k** *w1118* (control), *UAS-Alg3^RNAi^*, *UAS-Alg9^RNAi^*, and *UASp-Pngl* were overexpressed by *esg^ts^*. Flies were raised at 18 °C and then shifted to 29 °C for 4 days (**b**) or 6 days (**k**) before dissection. *Puc-lacZ^E69^* was used to indicate the activation of JNK signaling. Nuclei were labeled in blue. **f** Flies were raised at 18 °C. Two-day-old *esg^ts^*; *UAS-GFP* female files were shifted from 18 to 29 °C for 2 days and then treated with 5% sucrose (control), *Ecc15*, or *P.e.* for 18 h. RT-qPCR was performed on cDNA samples extracted from the FACS-sorted GFP⁺ progenitor cells. Quantification data represent the mean ± SD (*n* = 3 replicates/treatment/gene, two-tailed unpaired *t*-test, **P* = 0.0227, ***P* = 0.0077, ****P* = 0.0003, *****p* < 0.0001). **j** *UAS-grnd^WT^-V5* (wild-type Grnd) or *UAS-grnd^N63A^-V5* (*N*-glycosylation site mutation form Grnd) was overexpressed in progenitors using *esg^ts^* at 29 °C for 7 days. *esg^ts^ > w1118* flies were used as controls. Five midguts per sample were dissected for western blot analysis using anti-V5-/·tubulin antibodies for blotting. **l** The data presented in (**a**–**k**) collectively support a model in which ALG3/9-mediated *N*-glycosylation of Grnd functions as a pivotal switch for JNK activation in progenitor cells upon infection. Images in (**b**) are representative of three independent experiments. Images in (**j**, **k**) are representative of two independent experiments. Scale bars: 15 μm. Source data are provided as a Source Data file.

repression targets in ECs in addition to JNK signaling that affect ISC proliferation. These data indicate that ALG3 and ALG9 repress ISC proliferation, at least in part, by inhibiting JNK activation. Since *puc-lacZ^E69^* induction could be detected in progenitor cells of *Ecc15*-infected guts (Fig. 2c), we asked whether ALG3 and ALG9 are regulated by gut damage. qRT-PCR analyses on fluorescence-activated cell sorting (FACS)-sorted progenitor cells (GFP⁺), showed that both *Alg3* and *Alg9* mRNA levels were significantly reduced in progenitors by enteric *Ecc15* or *P.e.* infection (Fig. 5f). Consistent with this, our previous cell type-specific RNA-seq data revealed that *Alg3* and *Alg9* are expressed in all midgut cell types and that both are markedly decreased in EBs by *P.e.* infection[37]. Therefore, we suggest that the downregulation of *Alg3* and *Alg9* could activate JNK during the gut damage response. In line with

this, overexpressing *Alg3* in progenitors significantly suppressed damage-induced ISC hyperproliferation (Fig. 5g), confirming that *Alg* genes can act to restrain the proliferative response during infection stress.

Beyond the *Alg* genes, we also investigated whether *N*-glycosylation in general is important for regulating ISC proliferation. We found that progenitor-specific overexpression of *Pngl*, the *Drosophila* ortholog of peptide *N*-glycosidase (*PNGase*) that catalyzes the de-glycosylation of glycoproteins, resulted in increased ISC proliferation (Fig. 5h), whereas overexpression of *Pngl^C303A^*, a mutant form of *Pngl* that lacks the enzymatic activity[38], was not pro-mitotic (Fig. 5h). Similarly, overexpression of *mNgly1*, the mouse *PNGase* ortholog, also promoted ISC proliferation (Fig. 5h). These results

support our proposal that the *N*-glycosylation normally restricts ISC proliferation.

We next explored the potential mechanisms by which ALG-mediated *N*-glycosylation restricts ISC proliferation. Using *Drosophila* wing imaginal discs, de Vreede et al. found that Grnd is *N*-glycosylated at asparagine 63 (N63) by ALGs and that this modification limits Grnd's binding to Egr[12]. As shown above, overexpression of wild-type *grnd* in ISCs did not promote ISC proliferation (Fig. 3m). However, over-expression of *grnd*[N63A], a mutant form of *grnd* that is refractory to *N*-glycosylation, did stimulate ISC proliferation under precisely the same conditions (Fig. 5i). Furthermore, western blot data showed that progenitor-expressed wild-type Grnd has a higher molecular weight than Grnd[N63A] (Fig. 5j), consistent with Grnd being *N*-glycosylated at N63 in the fly midgut. In addition, ectopic expression of *Pngl* in progenitors activated JNK signaling in these cells, as indicated by *puc-lacZ*[E69] induction (Fig. 5k). These data corroborate our proposal that *N*-glycosylation of Grnd normally blocks the Grnd–Egr interaction and thereby inhibits the activation of JNK signaling. Upon infection, the downregulation of *Alg3* and *Alg9* may impair the *N*-glycosylation of Grnd, thereby increasing its interaction with Egr to facilitate the activation of JNK in both ECs and progenitor cells (Fig. 5l).

In addition to their role in ISC mitosis, ALG3 and ALG9 also appeared to regulate stem cell numbers in a non-autonomous manner. Notably, our observations revealed that the knockdown of *Alg3* in EBs led to a significant increase in stem cell numbers, as evidenced by an increase in the number of cells expressing the ISC marker *Dl-lacZ* (Supplementary Fig. 5a, b). Previous studies have highlighted that JNK signaling induces symmetric divisions in ISCs by reorienting the mitotic spindles[26]. Given that the pro-mitotic effect of *Alg3* depletion in EBs is reliant on Egr (Fig. 5d), we suggest that ALG3 and ALG9 in EBs may restrain ISC symmetric divisions via a non-cell-autonomous signaling involving the JNK pathway.

### JNK activation in ECs stimulates EGFR signaling in ISCs via Rhomboid

We next focused on how the JNK ligand Egr is regulated in the midgut. As shown above (Fig. 1), Egr expression is induced specifically in progenitors by aging and stress. To test whether this induction might be a secondary effect of ISC proliferation, we used an RNAi against *String* (*Stg*), a *Cdc25* homolog required for ISC mitoses[39,40]. As shown in Supplementary Fig. 6a, b, bacterial infection induced a striking upregulation of *egr* and massive ISC proliferation. RNAi-mediated knockdown of *stg* totally blocked *P.e.*-induced ISC mitoses (Supplementary Fig. 6b), but did not block the induction of *egr* (Supplementary Fig. 6a, c), indicating that the induction of Egr is not a consequence of ISC proliferation, and may be more directly controlled by signaling. As ECs serve as the primary sensors of damage[41], we first checked whether stress-dependent upregulation of Egr in progenitors correlates with mitogen induction in ECs. During stress-induced gut damage responses, two major signaling proteins, Yorkie (Yki, the downstream transcription activator of the Hippo pathway) and JNK, are turned on in ECs to generate mitogens (e.g., the cytokines Upd3, Upd2, and the neuregulin Krn, etc.) that support ISC proliferation[41]. Interestingly, over-expression of *yki* in ECs strongly induced Upd3 in ECs, but failed to induce *egr* expression in progenitors (Fig. 6a). Furthermore, whereas activation of JNK signaling in ECs by *puc*[RNAi] strongly upregulated both Upd3 in ECs and *egr* in progenitors (Fig. 6a), direct overexpression of *upd3* in ECs did not promote *egr* induction in progenitors (Fig. 6a). This suggested that signal(s) other than the Upd cytokines are involved. Our previous work indicated that Rho, an intermembrane protease that promotes the cleavage and secretion of EGFR ligands[42], could be upregulated in ECs by *Hep*[Act43]. Tellingly, overexpression of *grnd* or *puc*[RNAi] in ECs induced *rho* expression in these cells (Fig. 6b). In addition, ISC hyperproliferation caused by *grnd* overexpression in ECs was repressed by *rho* depletion (Fig. 6c). These data suggest that Rho is a downstream target of Grnd/JNK signaling. Consistently, overexpression of *rho* in ECs induced a striking upregulation of *egr* in progenitors (Fig. 6a). These findings indicate that the Grnd-JNK-Rho signaling in ECs plays a crucial role in regulating *egr* expression in progenitor cells.

Next, we tested the interaction of Rho with EGFR ligands. Four activating EGFR ligands (Spi, Krn, Gurken/Grk, and Vein/Vn) and one inhibitory ligand (Argos) have been identified in *Drosophila*[44]. Three of these (Spi, Krn, and Grk) are produced as transmembrane precursors and require cleavage by Rhomboid proteases to activate their secretion[44]. Grk has not been detected in the midgut[45], and Vn is produced in gut visceral muscle as a secreted protein that does not require Rho for activation[43]. In gain-of-function tests, overexpression of *rho* in ECs strongly induced ISC proliferation (Fig. 6d), had a weaker effect when expressed in ISCs (Supplementary Fig. 7a), and had no significant pro-mitotic activity when overexpressed in EBs or EEs (Supplementary Fig. 7b, c). Epistasis tests in which we co-expressed *rho* together with *Krn*[RNAi] and/or *spi*[RNAi] showed that Rho-driven ISC mitoses depended on both ligands, and that Rho potentially acts in both ECs and progenitor cells (Fig. 6e, f). These results support a model which Grnd-JNK-Rho-Krn/Spi signaling in ECs controls the activation of EGFR, and proliferation, in progenitor cells.

### Eiger is a downstream target of EGFR-MAPK signaling

In further tests, we found that overexpressing either *Krn* or a secreted variant of *Spi* (*sSpi*) in ECs was sufficient to strongly induce *egr* expression in progenitor cells (Fig. 6g, h). Given that EB- and EC-derived Spi and Krn activate EGFR/MAPK signaling in progenitor cells[40,43,45,46], we next examined whether *egr* is a target of the EGFR/MAPK pathway. Artificially activating MAPK/ERK signaling in progenitors by expressing *Ras*[V12S35] or *Raf*[GOF] strongly induced *egr* expression (Fig. 7a). Conversely, depletion of *Egfr* in progenitors totally blocked *P.e.*-induced *egr* induction (Fig.7c) as well as ISC hyperproliferation (Fig. 7b). Similarly, knockdown of *Ras* in progenitors totally blocked *egr* induction by *Ecc15* infection (Fig. 7d). These data indicate that Egr is a downstream target of EGFR/Ras/Raf signaling in progenitor cells. When considered together with our discovery that progenitor-derived Egr signals to Grnd in ECs, this implies that signaling between progenitor cells and ECs forms an Egr→Grnd→JNK→Rho→Krn/Spi→EGFR→Egr feedforward loop capable of maintaining and potentially amplifying JNK activity in ECs, and pro-proliferative EGFR signaling in progenitor cells (Fig. 7e).

Finally, we used a unique experimental condition to further validate this signaling interaction, by testing a condition in which Egr was expressed specifically in progenitors and Grnd was coincidently suppressed specifically in ECs. In a previous study, we identified an SH3PX1-dependent autophagy-endocytosis network that specifically controls EGFR degradation and activity in progenitor cells of fly midgut[40]. Homozygous *SH3PX1* mutant flies exhibit exclusive activation of EGFR in progenitors, leading to ISC hyperproliferation, while loss-of-function of *SH3PX1* in ECs and EEs has no impact on EGFR activity and ISC mitosis[40]. Thus the *SH3PX1* mutant effectively mimics specific activation of EGFR in progenitors. This is likely because EGFR is expressed preferentially in progenitors[40,47], such that blocking its degradation using the *SH3PX1* mutation causes EGFR accumulation and activation only in these cells. As expected, *SH3PX1*[d1/10A] mutants specifically induced *egr* expression in progenitors (Supplementary Fig. 8a), and concomitant knockdown of *egr* in these progenitors suppressed *SH3PX1*[RNAi]-induced ISC hyperproliferation (Supplementary Fig. 8b), confirming *egr* as the downstream target of SH3PX1. Hence, the *SH3PX1* mutant mimics the overexpression of Egr in progenitors. Importantly, depleting *grnd* in ECs effectively blocked the pro-mitotic phenotype in the *SH3PX1*[d1/HK62b] mutants (Supplementary Fig. 8c), consistent with our proposal that progenitor-derived Egr signals to Grnd in ECs to indirectly promote ISC proliferation. Altogether, this study uncovers a paracrine JNK-EGFR-JNK feedforward loop that sustains ISC proliferation during stress-induced gut

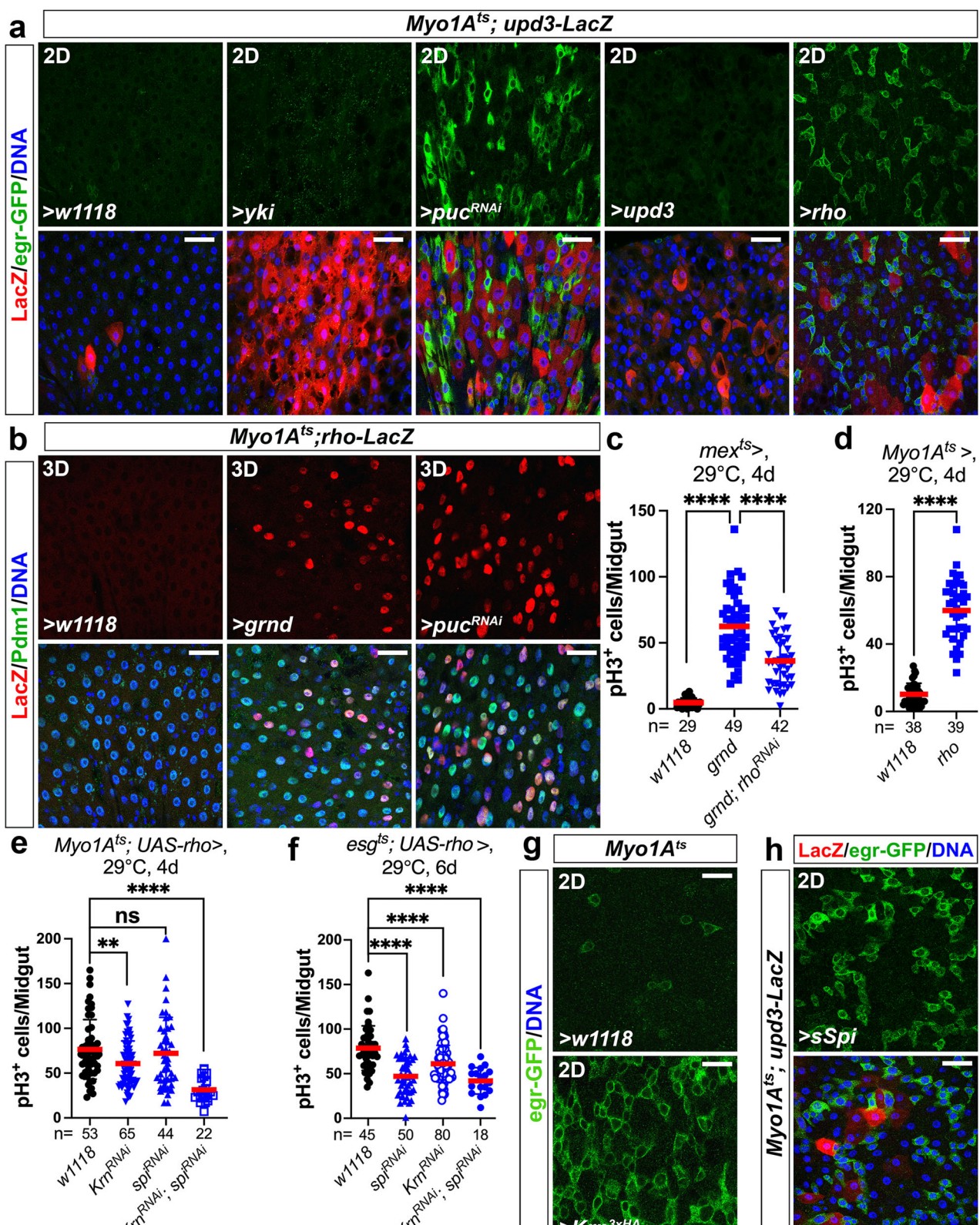

**Fig. 6 | JNK activation in ECs upregulates Rho.** Different genetic manipulations are indicated in individual panels. 2–3-day-old adult female flies were shifted from 18 to 29 °C for 2, 3, 4, or 6 days (as indicated in panels) before dissection. Midguts were stained with anti-GFP/-LacZ (**a**, **h**), anti-Pdm1/-LacZ (**b**), anti-pH3 (**c**–**f**), or anti-GFP (**g**) antibodies. Nuclei were labeled in blue. Induction of *egr*, *upd3*, or *rho* was indicated by *egr-GFP* (**a**, **g**, **h**), *upd3-lacZ* (**a**), or *rho-lacZ* (**b**), respectively. **c**–**f** ISC mitoses were quantified by counting pH3+ cells. Quantification data represent the mean ± SD (two-tailed unpaired *t*-test, ns*P* = 0.5529, **P* = 0.0041, ****P* < 0.0001). *N* values in individual panels indicate the number of midguts examined. Images in (**a**, **b**, **g**, **h**) are representative of three independent experiments. Scale bars: 30 µm in (**a**, **b**, **h**) and 15 µm in (**g**). Source data are provided as a Source Data file.

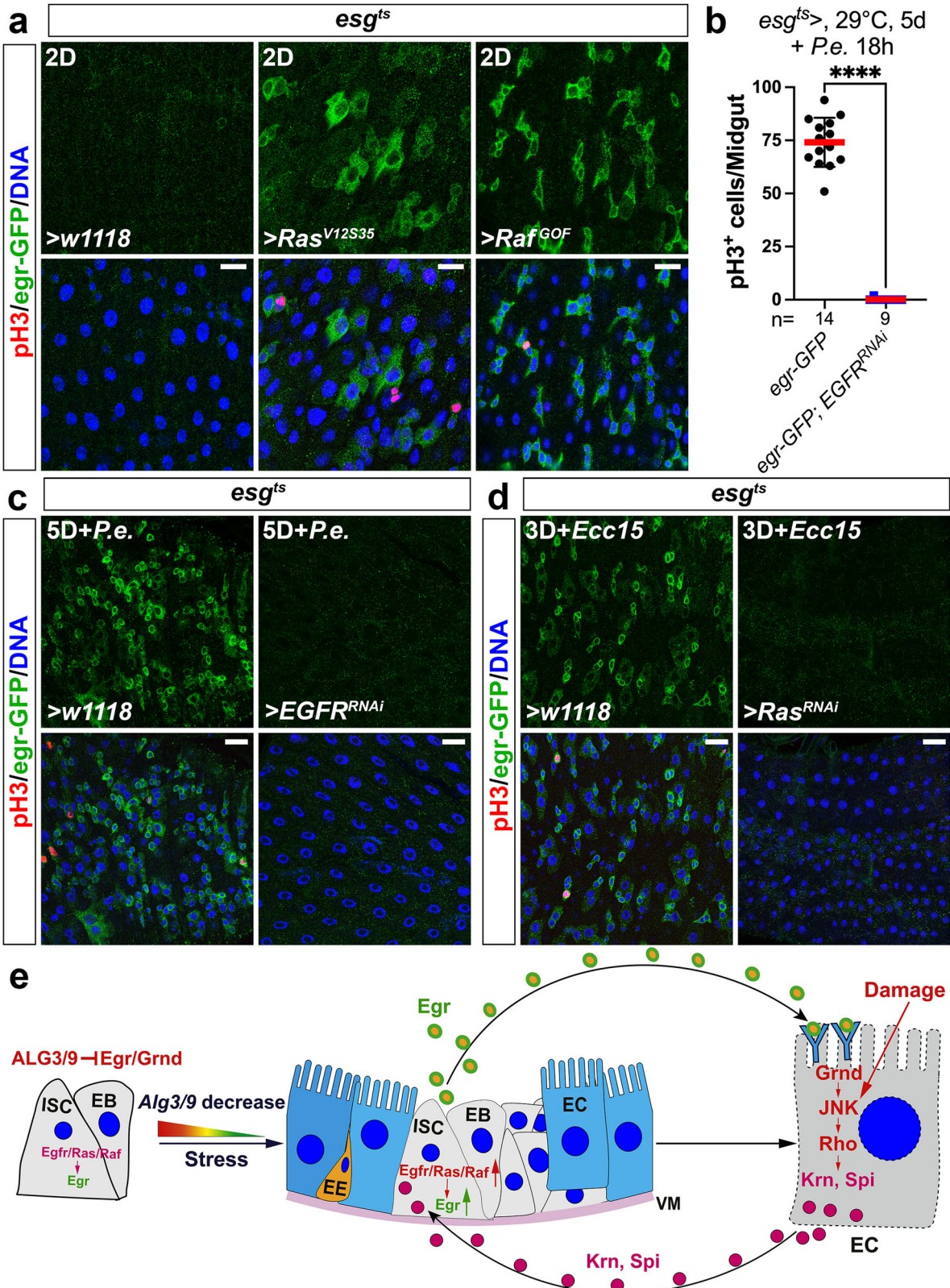

**Fig. 7 | Egr is a downstream target of EGFR-MAPK signaling. a–d** Various genetic manipulations in progenitors driven by *esg^ts* (**c**). 2–3-day-old adult females were shifted from 18 to 29 °C for 2, 3, or 5 days (as indicated in panels) before dissection or infection. Midguts were stained with anti-GFP/-pH3 antibodies. Nuclei were labeled in blue. **b** ISC mitoses were quantified by counting pH3⁺ cells. Quantification data represent the mean ± SD (two-tailed unpaired *t*-test, ****$P < 0.0001$), with each dot representing one sample. *N* values indicate the number of midguts examined. Images in (**a**, **c**, **d**) are representative of three independent experiments. Scale bars: 15 μm in (**a**) and 20 μm in (**c**, **d**). **e** The working model proposed in this paper illustrates a spatial activation pattern of JNK signaling within the intestinal stem cell lineage. Our findings reveal a previously unrecognized Egr/Grnd/JNK-Rho-Krn/Spi-EGFR-Egr pathway that establishes a feedforward loop between ECs and progenitor cells. This feedforward loop could be initiated either by stress-induced downregulation of ALG3 and ALG9 in progenitors or by damage that directly stimulates JNK signaling in ECs. It plays a pivotal role in maintaining Egr production in progenitors and sustaining JNK activation during gut epithelial regeneration. Source data are provided as a Source Data file.

regeneration, with ALG3 and ALG9 serving as required brakes to this pro-proliferative loop (Fig. 7e).

## Discussion

Tissue homeostasis depends on the precise control of resident stem cells, which is crucial in highly regenerative tissues like the gastrointestinal tract. In both humans and insects, the gut epithelium undergoes rapid turnover at rates influenced by factors such as diet, gut microbiota, age, and exposure to toxins and pathogens. Aging and environmental stresses can disrupt stem cell functions, leading to reduced regenerative capacity and increased susceptibility to age-related disorders, including stem cell loss and atrophy, or stem cell hyperactivity as in inflammatory bowel diseases (IBD) and gastrointestinal cancers. Addressing the mechanisms that govern stem cell functions in response to stress can provide valuable insights relevant to the prevention and treatment of such diseases. A prevalent working model of *Drosophila* gut regeneration is that ECs act as primary sensors of stress that deliver mitogenic signals to ISCs, which then divide to drive epithelial renewal[41,48]. However, how ISC-intrinsic factors might respond to damage-dependent signaling is less well understood. Recent studies have reported that calcium signaling can integrate a wide range of stress-based inputs to ISC proliferation[49,50], and that an endocytosis-autophagy network restrains ISC proliferation cell autonomously[40,51]. The function of this latter network declines during aging or stress, causing ISC hyperplasia[52,53]. These findings highlight the complex interplay of ISC-extrinsic and -intrinsic factors in governing gut regeneration and suggest therapeutic interventions that might address age-related intestinal disorders.

Over the last decade much progress has been made in understanding the roles of core JNKK cascade components in *Drosophila* gut homeostasis, but the specific functions of the JNK pathway ligand, the TNF ortholog Egr, and its two receptors, Wgn and Grnd, had not until now been clarified. Recent studies noted Egr's specific expression pattern in the gut progenitor cells[34,54], but its function as a damage- and aging-inducible factor had not been reported. In this study, we also report that Grnd, rather than Wgn, is the major functional receptor for Egr in the gut stem cell lineage, at least for the proliferative response. This is consistent with the recent publication of Loudhaief et al., showing that Wgn's primary function in the gut is to regulate lipid metabolism in ECs, a function independent of Egr[55]. Perhaps more importantly, we also report a novel activation pattern of the JNK pathway in the gut, wherein Egr is exclusively expressed in progenitor cells but does not activate Grnd or trigger JNK signaling in these cells. Instead, Egr acts in a paracrine manner to activate JNK signaling in adjacent, post-mitotic differentiated ECs. The mechanisms that underlie the refractivity of progenitor cells to Egr reception and JNK activation remain unclear and will be interesting topics for future study. Regardless, the spatial partitioning of JNK signaling appears to be important in gut physiology because high levels of JNK activity in progenitor cells cause stem cell loss by apoptosis. In contrast (and as described below) JNK activation in differentiated ECs initiates a signaling cascade that promotes gut epithelial regeneration mediated by stem cell proliferation.

In this study, we also discovered that *N*-linked glycosylation pathway genes provide an ISC-intrinsic restraint that controls ISC proliferation. Protein *N*-glycosylation at asparagines is a highly conserved process in eukaryotes, and its malfunction has been associated with a wide range of human diseases[56]. Through genetic screening, we identified the mannosyltransferases ALG3 and ALG9, which add mannosyl groups to target proteins during *N*-glycosylation, as critical restraints on ISC proliferation. Interestingly, the other three ALGs (ALG2, ALG11, and ALG12), which are supposed to have similar functions to ALG3 and ALG9, did not exhibit obvious anti-mitotic roles in ISCs. The reason for this remains unclear. Our genetic data implied that the JNK pathway receptor, Grnd, is likely an important target of ALG3 and ALG9 in midgut cells, as previously shown by de Vreede et al.[12] in

*Drosophila* wing imaginal discs. Our results suggest that stress-induced downregulation of *Alg3* and/or *Alg9* might be one trigger that promotes Grnd–Egr interaction to stimulate ISC divisions (Fig. 5l). The mechanism underlying the damage-dependent downregulation of ALG function in progenitor cells remains to be determined.

We also addressed the mechanisms of induction of Egr in progenitor cells following gut damage. Surprisingly, our findings indicated that the activation of JNK in ECs is, indirectly, responsible for Egr induction in progenitors. Mechanistically, JNK activation upregulates the intermembrane protease Rho in ECs, and our data suggest that Rho then promotes the secretion of Krn and Spi, which in turn activate EGFR signaling in progenitors to stimulate the production of more Egr. This feedforward (i.e., positive feedback) loop could be initiated either by stress-induced downregulation of ALG3 and ALG9 in progenitors, or by damage that directly stimulates JNK signaling in ECs (Fig. 7e). The mechanistic details of this feedforward loop between cell types were previously unknown for the fly gut and may be relevant to other systems in which spikes in JNK and EGFR signaling coincide and synergize. In the fly gut, the reciprocal activation relationship between JNK and EGFR signaling plays a pivotal role in regulating ISC proliferation and maintaining gut homeostasis, and a similar relationship could be at play in other regenerative tissues. Limiting and terminating this feedforward reaction at the close of a regenerative episode is obviously also important, and how this works is an interesting topic for future studies. A recent study reported that several micro-RNAs are upregulated at the termination stage of gut regeneration, and required for the process[57], but their potential action as suppressors of JNK and EGFR signaling is not yet known. The JNK downstream target Puc probably also plays a role in the termination of gut regeneration. Since Puc, a Jun-kinase-phosphatase, acts as a negative feedback inhibitor of the JNK pathway, once JNK is activated in ECs, Puc accumulates over time. Consequently, Puc can suppress JNK activity, potentially leading to decreases in the downstream targets of JNK, including Rho, secreted Spi and Krn, and EGFR's pro-proliferative targets[58,59].

Notwithstanding the central role of Upd cytokines in intestinal regeneration, genetic tests showed that depletion of Upd2 and Upd3 in ECs only partially suppressed stress responses and regeneration, suggesting the involvement of other mitogens[41]. Indeed the Rho-controlled EGFR ligands are well known as powerful ISC mitogens[40,43,46,58,59]. Rho can be induced in a cell-autonomous fashion by epithelial stress[40] and/or the loss of E-cadherin in apoptotic ECs[46], but as we show here, Rho in ECs also responds to signaling from other cells (e.g., Egr from ISCs/EBs). The precise mechanism by which Egr–Grnd–JNK signaling upregulates Rho, whether through the canonical AP-1 transcription factor or the recently defined JNK effector, Ets21c[60], requires further investigation.

In certain cancers, interactions between the JNK and EGFR pathways mediate a switch in JNK's function from pro-apoptotic to pro-tumorigenic[13]. The feedforward loop between JNK and EGFR that we identify here provides a specific mechanism that may be relevant to JNK's pro-tumorigenic functions. Inhibitors of TNF/TNFR signaling are extensively used to treat inflammatory diseases such as ulcerative colitis and rheumatoid arthritis[61] and EGFR inhibitors are now used in many anti-cancer therapies[62]. However, both therapeutic approaches face challenges due to variable efficacy and rapidly evolved resistance. Based on the mutual amplification mechanism identified in this study, between JNK and EGFR signaling, combination therapies involving both TNF/TNFR and EGFR inhibitors deserve special attention.

## Methods

### Fly stocks

*UAS-egr[w]* and *UAS-egr[reg1]* were obtained from Tian Xu (Westlake University, China). *UAS-grnd[WT]-V5* and *UAS-grnd[N63A]-V5* were obtained from David Bilder (UC Berkeley, USA). *Puc-lacZ[E69]* was obtained from Jin Jiang (UT Southwestern Medical Center, USA). *UAS-wgn* was obtained

from Julien Colombani (University of Copenhagen, Denmark). *SH3PX1^{10A}* was obtained from Kate O'Connor-Giles (University of Wisconsin-Madison, USA)[63]. *SH3PX1^{HK62b}* was obtained from Helene Knævelsrud (Oslo University Hospital, Norway)[64]. The following stocks were made in this study: *Hml-Gal4/CyO; tub-Gal80^{ts}, UAS-GFP^{NLS}/TM6B* (*Hml^{ts}*), *tub-Gal80^{ts}, UAS-GFP^{NLS}/CyO; Lpp-Gal4/TM6B* (*Lpp^{ts}*); *mex-Gal4/ CyO; tub-Gal80^{ts}, UAS-GFP^{NLS}/TM6B* (*mex^{ts}*), *Myo1A-Gal4, tub-Gal80^{ts}/ CyO; upd3-lacZ/TM6B*, and the "Su(H)GBE^{ts}-FlipOut" (*Su(H)GBE^{ts} F/O*) lineage-tracing system (genotype: *Su(H)GBE-Gal4, UAS-GFP, tub-Gal80^{ts}/CyO; UAS-flp,act > CD2 > Gal4/TM6B*). The following stocks were obtained from the Edgar Lab stock collections (University of Utah, USA): *tub-Gal80^{ts}/CyO; Dl-Gal4/TM6B* (*Dl^{ts}*), *esg-Gal4, UAS-GFP/ CyO; tub-Gal80^{ts}/TM6B* (*esg^{ts}*#1), *esg-Gal4, UAS-GFP, tub-Gal80^{ts}/CyO* (*esg^{ts}*#2), *tub-Gal80^{ts}/FM7; esg-Gal4/CyO* (*esg^{ts}*#3), *Su(H)GBE-Gal4, UAS-GFP/CyO; ubi-Gal80^{ts}/TM6B* (*Su(H)GBE^{ts}*), *Myo1A-Gal4, tub-Gal80^{ts}, UAS-GFP/CyO* (*Myo1A^{ts}*), *tub-Gal80^{ts}, UAS-GFP/CyO; prosV1-Gal4/TM6B* (*prosV1^{ts}*), *w1118, esg-lacZ, UAS-Hep^{Act}, UAS-P3S; UAS-Hep^{Act}, upd3-lacZ, rho-lacZ^{x81}, Dl-lacZ, UAS-yki, UAS-upd3, UAS-puc^{RNAi}, UAS-rho, UAS-Ras^{V12S35}, UAS-Raf^{GOF}*, and *SH3PX1^{d1}*. *UAS-grnd^{RNAi}* (104538) and *UAS-Alg11^{RNAi}* (104286) were obtained from the Vienna *Drosophila* RNAi Center. *UASp-Pngl* (109-620), *UASp-Pngl^{C303A}* (109-623), and *UASp-mNgly1* (109-626) were obtained from the Kyoto *Drosophila* Stock Center. *UAS-Krn^{3xHA}* (F002754) was obtained from the FlyORF collections. *UAS-stg^{RNAi}* (1395R-1) and *UAS-Krn^{RNAi}* (8056R-3) were obtained from the National Institute of Genetics (NIG, Japan). The following stocks were obtained from the Bloomington *Drosophila* Stock Center: *eiger-GFP* (66381), *UAS-egr^{RNAi#1}* (55276), *UAS-egr^{RNAi#2}* (58993), *UAS-wgn^{RNAi}* (58994), *UAS-P35* (6298), *UAS-Alg2^{RNAi}* (55671), *UAS-Alg3^{RNAi}* (53350), *UAS-Alg9^{RNAi}* (65998), *UAS-Alg12^{RNAi}* (34680), *UAS-Bsk^{K53R}* (9311), *UAS-Alg3* (90943), *TRE-DsRed* (59011), *UAS-Egfr^{RNAi}* (25781), *UAS-Ras^{RNAi}* (29319), *UAS-Rho^{RNAi}* (28690), *UAS-spi^{RNAi}* (28387), *UAS-sSpi* (58436), *UAS-SH3PX1^{RNAi}* (27653), and *G-TRACE* (28281).

## Bacteria and bleomycin treatments
*P.e.* and *Ecc15* were obtained from Nicolas Buchon (Cornell University, USA). They were cultured in LB medium (1 mL glycerol stock in 200 mL LB) containing Rifampicin (100 µg/mL) at 29 °C with shaking at 190 rpm for 27 h. The culture was centrifuged at 2800 × g for 20 min at 4 °C, and the resulting pellet was resuspended in 8 mL 5% sucrose for fly enteric infection. Bleomycin was diluted with 5% sucrose to a final concentration of 250 or 500 µM for fly feeding, as specified in the corresponding figure legends.

## Immunostaining
After dissection, samples were fixed in a solution of PBS containing 8% paraformaldehyde for 30 min, washed in PBS with 0.1% Triton X-100, and blocked in PBS with 0.1% Triton X-100 and 10% normal goat serum for at least 30 min at room temperature. Following blocking, all samples were incubated with primary antibodies at 4 °C overnight using the following dilutions: chicken α-GFP (Thermo Fisher Scientific, #A10262, 1:1000), rabbit α-pH3 (Millipore, # 06-570, 1:1000), mouse α-β-galactosidase (Promega, #Z3781, 1:300), rabbit α-cleaved *Drosophila* Dcp-1 (Asp215) (Cell Signaling Technology, #9578, 1:100), rabbit α-Pdm1 (gift from X. Yang at Zhejiang University, China, 1:200), rat α-HA (Roche, # 11867423001, 1:300), rabbit α-DsRed (Takara Bio Clontech, #632496, 1:400), mouse α-Armadillo (DSHB, #N2 7A1, 1:10). Staining signals were detected using species-appropriate secondary antibodies conjugated with Alexa Fluor 488, and/or 568, and/or 633. (Thermo Fisher Scientific, 1:1000). Nuclei were labeled using DAPI (Thermo Fisher Scientific, #D1306, 1:1000) or Hoechst 33342 (Thermo Fisher Scientific, #62249, 1:1000).

## Western blots
Guts from female flies (5 guts/tube) are dissected and placed in cold PBS in 1.7 ml Eppendorf tubes on ice. A short pulse of centrifugation draws the dissected gut samples to the bottom of the tube, and any excess PBS is removed. The gut samples are snap-frozen by immersing the bottom of the tubes in a dry ice/ethanol bath. To prevent degradation by gut proteases, the rapid permeation of SDS is optimized: the frozen guts are rapidly thawed by adding 50 µl of 2× sample buffer per sample. Tubes are immediately cap-locked and placed in an Eppendorf Thermomixer (preheated to 100 °C) for a 10-min incubation at 2000 rpm. After incubation, lysates are spun for 10 min at 13,000 rpm at 4 °C. Supernatants are collected and run on SDS–PAGE for western blot analysis (20 µl/lane). The following antibodies are used for blotting: mouse α-tubulin (DSHB, #12G10, 1:250) and mouse α-V5 (Invitrogen, #46-0705, 1:2000). Preparation of 2× sample buffer (10 ml): 50% glycerol 2.4 ml, 10% SDS 2 ml, 1 M Tris (pH 6.8) 0.8 ml, β-mercaptoethanol 1.0 ml, ddH$_2$O 3.8 ml.

## GFP intensity quantification from fluorescent microscopy images
For all quantifications of GFP intensity shown in Fig. 4d, e, the DAPI channel of each image was segmented using ImageJ. First, background fluorescence was removed via no-neighbor de-blurring, then images were thresholded, and a watershed algorithm was applied. After segmentation, the mean GFP fluorescence intensity was measured for all objects (e.g., nuclei) with a minimum circularity of 0.5. To calculate the frequency of progenitors marked by GFP expression, GFP fluorescence measurements from individual cells from the *w1118* control samples were thresholded using Otsu's algorithm. Clearly visible cells with fluorescence levels above the threshold and a nuclear size <25 µm$^2$ were then marked as progenitors.

## FACS sorting and RT-qPCR
Midguts were dissected in cold nuclease-free PBS and digested with 5 mg/mL collagenase for 1 h at 29 °C in a thermomixer (850 rpm). Samples were gently pipetted every 15 min to aid in dissociation. After digestion, samples were centrifuged at 600 g for 10 min, resuspended in cold nuclease-free PBS, and filtered into a 35 µm strainer cap FACS tube. The BD FACSAria cell sorter was employed to collect the GFP-positive single progenitors (~20,000 cells/sample) for subsequent RT-qPCR. RNA was extracted from FACS-sorted progenitors using the Arcturus picoPure RNA Isolation Kit. For each replicate and condition, cDNA was synthesized from 100 ng total RNA using the NEB ProtoScript® II First Strand cDNA Synthesis Kit. qPCR analyses were performed using the Bio-Rad SYBR Green PCR kit and the CFX384TM Real-Time System (Bio-Rad). Data collection involved three independent biological replicates, with error bars indicating the measurement range. Housekeeping gene, *β-Tubulin at 56D* (*βTub56D*), was used as the internal reference for target gene expression. The ΔΔCt method was used to calculate the relative gene expression. Primer sequences used were:

> *Alg3* forward: CTGTACTCGAAGAGCAGAAAGG
> *Alg3* reverse: ACAGTCGCAGCACGTATATC
> *Alg9* forward: CGGCTCTGGAACCCAATTAT
> *Alg9* reverse: GGTCAGGAATGGCACATAGAA
> *βTub56D* forward: ACATCCCGCCCCGTGGTC
> *βTub56D* reverse: AGAAAGCCTTGCGCCTGAACATAG

## Statistics and reproducibility
Statistical analyses were conducted using the GraphPad Prism 10 software package. Statistical significance was denoted as follows: non-significant (ns) $P > 0.05$, $*P < 0.05$, $**P < 0.01$, $***P < 0.001$, and $****P < 0.0001$. All representative images were independently replicated at least 2–3 times with consistent results obtained.

## Reporting summary
Further information on research design is available in the Nature Portfolio Reporting Summary linked to this article.

## Data availability

All data generated in this study are provided in the main figures, Supplementary Information, and the Source data file. The raw confocal microscopy images supporting the findings of this study are available from the corresponding authors upon request. Source data are provided with this paper.

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

## Acknowledgements

We thank David Bilder, Tian Xu, Laura Johnston, Jin Jiang, Julien Colombani, Xiaohang Yang, Kate O'Connor-Giles, Helene Knævelsrud, and Nicolas Buchon for providing fly stocks, bacteria, and antibodies; Clothilde Penalva for gut RNA-seq data. We also thank the Cell Imaging Core and the Flow Cytometry Core at the University of Utah Health Sciences Center for their support. This work was supported by the Huntsman Cancer Foundation and grants from the National Institutes of Health (R01 GM124434, R35 GM140900 to BAE, and P30 CA042014).

## Author contributions

Conceptualization was provided by P.Z. and B.A.E. Methodology was designed by P.Z. and B.A.E. Investigations were carried out by P.Z., S.M.P., C.Z., X.K., T.K., and C.L. Data analyses were performed by P.Z. and M.M. P.Z. and B.A.E. wrote the manuscript. All authors reviewed the manuscript.

## Competing interests

The authors declare no competing interests.
