## [Peer Review File · Nature Communications]

Inter-cell type interactions that control JNK signaling in the Drosophila intestineReviewer #1 (Remarks to the Author):

This manuscript by Zhang et al presents a new intercellular signaling crosstalk that sheds light into the mechanisms by which pathogenic damage induces JNK activation in intestinal epithelial cells (ECs) and how this stress signal from ECs leads to activation of intestinal stem cell (ISC) proliferation during intestinal regeneration. The authors propose that the upregulation of the TNF ligand Eiger (Egr) in ISC/EBs following damage signals to its receptor Grnd in ECs leading to JNK activation, the upregulation of the EGF processing protease Rhomboid (Rho) and subsequent secretion of EGF ligands from ECs. EC derived EGF ligands then activate EGFR signaling in ISCs/EBs to drive regenerative proliferation. Interestingly, the authors identify ALG3, a glycosylating enzyme, as responsible for the inactivation of Grnd in ISCs/EBs, making these cells refractive to the action of Egr/TNF. This work proposes mechanisms by which global damage in the intestine leads to the activation of stress sensing responses, such as JNK activation, largely exclusively in ECs and how this stress signal is relayed to ISCs to induce their proliferation. Overall, the data presented in this paper provides insights into the cellular and molecular underpinnings of long-standing unexplained phenomenology/observations in the field of intestinal regeneration in *Drosophila*. This is a valuable contribution to the field. However, there are several important observations made in this study that do not support the proposed model and conclusions made and for which no explanation is provided. Furthermore, some of the central conclusions made in the study do require additional data to be substantiated.

Major comments:

- Line 362: 'Egr is exclusively expressed in progenitor cells but does not activate Grnd or JNK in progenitors and instead does it only through activation of Grnd in ECs'. This statement represents a core component of the model presented by the authors. However, this has not been proven in any way. In fact, the impairment of infection induced ISC proliferation upon ISC/EB grnd knockdown (Figure 3F, I) directly contrasts with this statement. To unambiguously support the proposed working model of intercellular signaling, a dual driver genetic system that allows the independent manipulation of Egr and Grn in the two cell types involved is essential. For instance, upregulation of Egr in Bleo treated guts using the *lexA/lexAop* system in ISCs/EBs combined with EC loss of function of Grn. Assessment of the outcome on ISC proliferation here versus that resulting from concomitant stem/progenitor Egr overexpression and grnd knockdown in control and Bleo treated guts would allow to test the proposed model.
- Line 165: 'wgn depletion in any cell type does not suppress ISC proliferation'. In fact, wgn knockdown in all cases shown has effects on ISCs that oppose those of grnd knockdown with more proliferation observed. In the former case. Why and is this Egr dependent?
- The use of Delta-gal4 as an ISC driver is problematic. This driver is well known for being patchy and, as stated by the authors themselves, weak when compared to the more robust and widely used *esg-gal4*, *Su(H)-gal80*, which is used in a few instances in the paper. Not sure why it has not been used consistently, instead of Delta-gal4. Related to this, the lack of phenotypes when using the Delta-gal4 (e.g. following grnd overexpression or knockdown) are difficult to interpret and need to be re-assessed using *esg-gal4*, *Su(H)-gal80*.
- Lines 181-190: The rational and experimental design presented here is confusing as it is the main conclusion drawn from the results. Overexpression of grnd leading to JNK activation in ECs shows a non-cell autonomous and likely indirect effect of JNK activation by grnd, which does not relate to the intercellular signaling proposed in the study of cell autonomous activation of JNK by EC expressed Grnd.
- Lines 204-205: Assessment of EC renewal following JNK activation in ISCs/EBs (Figure 4c-d). How was this experiment done? Lineage tracing is necessary for this type of analysis.
- Figure 3n: Why is Grnd positioned in ISCs and not EBs where a function for the receptor has been shown in regeneration. This is also related to my first point.
- Figure 5: Role of Alg3 in the restriction of JNK activation.
 - a-Alg3 is downregulated in ISCs/EBs following Pe or Ecc15 infection. How do you reconcile this with the proposed model of JNK activation being inhibited by Alg3 in ISCs/EBs given that JNK is not activated (especially in Pe) in these cells during infection in spite of Alg3 being down?
 - b-Also, if Alg3 was the determinant factor as to whether JNK is activated or not downstream of Grnd, one would expect EC will be devoided of Alg3. Is this the case? What happens if Alg3 is

downregulated in ECs? Does this lead to JNK activation in homeostasis?

c- The data in Figure 5b shows that alg knockdown in progenitors also leads to predominant activation of JNK in EC. What does this mean? Overall, conclusions drawn from this data suffer from lack of experimental support.

Minor comments:

- Line 139: reference to previous work on haemocytes as a source Egr (30). The reference cited is not correct or at least not the first account of Egr in haemocytes, which came from Marcos Vidal's lab (Cordero et al, 2010-Dev Cell).

Reviewer #2 (Remarks to the Author):

In this manuscript, Zhang, Edgar and colleagues present a novel mechanism responsible for the increased proliferation of intestinal stem cells (ISCs) during aging and in the event of gut infection. They discovered that Egr (the Drosophila TNF), which is the ligand for the JNK signaling pathway, is specifically induced in intestinal progenitor cells as a result of aging or pathogenic infection. Egr then preferentially activates JNK signaling in the differentiated enterocytes by binding to the Grnd receptor, leading to the upregulation of Rhomboid expression. This triggers the secretion of two growth factor receptor (GFR) ligands, Krn and Spi, which stimulate ISC proliferation through EGFR/Ras signaling. Interestingly, the study reveals that the expression of Egr in ISCs is promoted by EGFR/Ras signaling.

Based on their findings, the authors propose the existence of a positive feedback loop involving JNK-EGFR signaling between ISCs and enterocytes, which drives ISC proliferation in the aging or damaged gut. The authors also demonstrate that the N-glycosylation of Grnd, facilitated by Alg3 and Alg9 enzymes in ISCs, prevents autonomous JNK signaling activation within the ISCs. In the context of an infected gut, however, the downregulation of Alg3 enables JNK signaling activation in ISCs.

The manuscript is skillfully written, and overall, the mechanistic study is thorough and logically organized. Given the common occurrence of dysregulated TNF/TNFR and EGFR signaling in inflammatory diseases and cancer, the present findings hold significant implications for disease understanding and the advancement of potential treatments. Nevertheless, certain conclusions drawn from the data presented lack sufficient support and should be adequately addressed during the revision process.

Major points:

1. The authors propose that N-glycosylation of Grnd specifically occurs in ISCs. To confirm this, it is necessary to determine the expression pattern of Grnd and N-glycosylated Grnd in the intestinal epithelium.
2. The authors highlight the significance of the expression level of alg3 in regulating the specific pattern of N-glycosylated Grnd. It is important to investigate whether alg3 is specifically expressed in ISCs and whether this pattern is disrupted in the infected gut. Also, is alg3 overexpression able to inhibit JNK activation in ISCs of the infected gut?
3. While Rhomboid is known to process EGFR ligand precursors, it is difficult to imagine to this reviewer why Rho overexpression in ECs has such a profound impact on ISC proliferation, especially considering the typically low expression of spi and krn in ECs in normal gut. Hence, it is worth investigating whether Rho overexpression in ECs causes transcriptional activation of spi and krn.
4. Can JNK activation or rho overexpression in ECs trigger cell apoptosis, or is it dose-dependent?

Minor points:

1. Figure 2, to demonstrate JNK activity also occurs in EEs, co-staining with an EE specific maker is necessary.
2. Figure 4d, which cell type is positive for Dcp-1 staining, ISCs or ECs?
3. Please describe the nature of the egr alleles: egrw and egrregg1
4. Could the authors provide speculation on how the positive feedback loop is halted during the recovery from infection, allowing the event to cease?

Reviewer #3 (Remarks to the Author):

Drosophila intestinal stem cells (ISCs) represent a powerful model to elucidate the mechanisms controlling and executing regeneration fueled by somatic stem cells. This manuscript describes research that significantly extends a large body of already published information on the pathways that relay stress signals to the stem cells to initiate and control tissue repair.

It had been recognized for a while that JNK and Ras/Erk signaling are involved in the activation of ISCs in response to stress or tissue damage. The work described by Bruce Edgar and colleagues clarifies how the two signaling systems intersect. The JNK pathway is activated in the terminally differentiated enterocytes by the TNF protein Eiger secreted from adjacent ISCs or enteroblasts. This in turn causes the activation of the protease Rhomboid and the release of EGFR ligands back to the ISCs which triggers them to initiate proliferation and differentiation resulting in repair or tissue damage. This feed forward loop can be modulated by Alg3-mediated glycosylation which interferes with the productive interaction of Eiger with its receptor Grindelwald on ISCs.

The new information reported here is well supported by the experimental evidence. It represents a significant gain in the understanding of somatic stem cell-driven tissue repair and regeneration in the fly gut. The described paracrine feedback mechanism may operate similarly in other examples of stem cell function and, more generally, in diverse examples of physiological or pathological cell-cell interaction.

General interest is high. Publication as a Nature communication paper is appropriate.

Minor comment:

Line 238:

"...expression of BskK53R (a dominant-negative form Bsk/JNK) significantly blocked ISC mitoses caused by Alg9RNAi (Fig. 5d)." That is not shown in fig 5d. And the statement seems to be contradicted in the next sentence.

Response to Reviewers (NCOMMS-23-42913-T)

We thank the reviewers for their positive assessments and constructive comments on our paper. All reviewers considered the work to be of significant interest and the presented data to be convincing, but listed a few concerns. We have carefully addressed each of the reviewers' comments through editorial revisions and by providing new data from additional experiments. Below, you will find our point-by-point response (in blue) to each of their comments.

Reviewer #1:

This manuscript by Zhang et al presents a new intercellular signaling crosstalk that sheds light into the mechanisms by which pathogenic damage induces JNK activation in intestinal epithelial cells (ECs) and how this stress signal from ECs leads to activation of intestinal stem cell (ISC) proliferation during intestinal regeneration. The authors propose that the upregulation of the TNF ligand Eiger (Egr) in ISC/EBs following damage signals to its receptor Grnd in ECs leading to JNK activation, the upregulation of the EGF processing protease Rhomboid (Rho) and subsequent secretion of EGF ligands from ECs. EC derived EGF ligands then activate EGFR signaling in ISCs/EBs to drive regenerative proliferation. Interestingly, the authors identify ALG3, a glycosylating enzyme, as responsible for the inactivation of Grnd in ISCs/EBs, making these cells refractive to the action of Egr/TNF. This work proposes mechanisms by which global damage in the intestine leads to the activation of stress sensing responses, such as JNK activation, largely exclusively in ECs and how this stress signal is relayed to ISCs to induce their proliferation. Overall, the data presented in this paper provides insights into the cellular and molecular underpinnings of long-standing unexplained phenomenology/observations in the field of intestinal regeneration in *Drosophila*. This is a valuable contribution to the field. However, there are several important observations made in this study that do not support the proposed model and conclusions made for which no explanation is provided. Furthermore, some of the central conclusions made in the study do require additional data to be substantiated.

Response: We thank the reviewer for the positive reception of our findings and the insightful criticism. In this revision, we have addressed the reviewer's comments with editorial revisions and new data from additional experiments, further strengthening our working model. Below, please find our point-by-point response (blue color) to each comment.

- Line 362: 'Egr is exclusively expressed in progenitor cells but does not activate Grnd or JNK in progenitors and instead does it only through activation of Grnd in ECs'. This statement represents a core component of the model presented by the authors. However, this has not been proven in any way. In fact, the impairment of infection induced ISC proliferation upon ISC/EB grnd knockdown (Figure 3F, I) directly contrasts with this statement.

Response: While we appreciate this reasonable concern, we respectfully disagree with the assessment that our model has not been proven in any way. We have provided multiple lines of evidence that strongly support our model:

(1) Our data showed that Egr is exclusively expressed in progenitor cells (Fig. 1), while JNK activation, indicated by its reporters *puc-lacZ* and *TRE-DsRed*, is predominantly observed in ECs

under both normal (Fig. 2a-2b and Supplementary Fig. S1) and regenerative conditions (Fig. 2c). Even upon overexpressing *grnd* in progenitors, *puc-lacZ* induction is still predominantly observed in ECs (Fig. 3o). These observations indicate that progenitor-derived Eiger signals to ECs for Grnd/JNK signaling activation via a cell non-autonomous (paracrine) mechanism.

(2) Our data in Fig. 3g-j and Supplementary Fig. S3c provide further support for this, by showing that Grnd loss from EBs, ECs, and the EB-EC lineage, but not from ISCs, suppresses stress-dependent stem cell proliferation. In Fig. 3d, k, m, n, S3d, and 6c we see the converse, namely the *grnd* overexpression in EBs or ECs but not ISCs stimulates ISC proliferation. These findings are consistent with the observed expression patterns of *puc-lacZ* (Fig. 2b) and *TRE-DsRed* (Supplementary Fig. S1), and with our model that Egr from ISCs activates Grnd in EBs and ECs.

We understand the reviewer's point of confusion regarding the observation that *grnd* overexpression in EBs triggers ISC mitoses, similar to its effects in ECs. This could be puzzling, considering that JNK activation is normally silenced in EBs, as indicated by *puc-lacZ*'s pattern. However, when considered in the context of dynamic gut regeneration, the phenomenon can be better understood. EBs are transient in nature and rapidly differentiate into ECs, and so overexpression of a gene in EBs often elicits similar effects as when it is overexpressed in ECs. We believe this is because newborn ECs inherit gene products produced in EBs. This phenomenon is not uncommon and has been observed with many gene manipulations, and the inheritance is evident when one tracks markers such as GFP. In this revision, we have updated the relevant results section (lines 188-194) to provide clearer support for our model, which is based in part on this cellular inheritance phenomenon.

To unambiguously support the proposed working model of intercellular signaling, a dual driver genetic system that allows the independent manipulation of Egr and Grnd in the two cell types involved is essential. For instance, upregulation of Egr in Bleo treated guts using the *lexA/lexAop* system in ISCs/EBs combined with EC loss of function of Grnd. Assessment of the outcome on ISC proliferation here versus that resulting from concomitant stem/progenitor Egr overexpression and *grnd* knockdown in control and Bleo treated guts would allow to test the proposed model.

Response: We agree that using the *Gal4/Gal80-LexA/LexAop* double system would be an attractive approach for rigorously testing the cell type-specific actions of Eiger and Grnd. However, for several reasons, this was not practical:

(1) For such genetic epistasis test, knocking down *grnd* in ECs first and then overexpressing *egr* in progenitors would be the ideal way, because RNAi typically takes 2-3 days to work whereas overexpression is effective within several hours. However, the only available *esg-LexA::GAD^{ts}* tool verified in the gut system relies on Gal80^{1,2}, just as do the conditional Gal4 drivers we use to target ECs (*Myo1A-Gal4*, *mex-Gal4*). This makes it impossible to separate the temporal expression phases of *egr* overexpression and *grnd*^{RNAi}, which would be necessary to silence *grnd* before *egr* was overexpressed. Hence, because we think this experiment would not work, we opted not to build the stocks for it.

(2) Additionally, the *LexAop-egr* fly that would be needed for this assay is not available, and making this new transgenic would take several months. Moreover, generating the stocks required

for the experiment would require several recombination steps to place different transgenes on the same chromosome, making this a very time-consuming experiment. There are potentially other ways to do the type of experiment suggested by the reviewer, and it is clearly an elegant approach. But all the options we could think of would require new transgenic constructions and complex, time-consuming crossing schemes that would probably take upwards of a year to accomplish. Moreover, they would use un-tested genetic approaches that might not work. Hence, we felt going this route was beyond what is needed for this paper, despite its theoretical advantages.

However, we did find an alternative solution that addresses this question, albeit somewhat indirectly. In a previous study, we identified an SH3PX1-dependent autophagy-endocytosis network that specifically controls EGFR degradation and activity in progenitor cells of the fly midgut³. Homozygous *SH3PX1* mutant flies exhibit activation of EGFR exclusively in progenitors, leading to ISC hyperproliferation, while loss-of-function of *SH3PX1* in ECs and EEs has no impact on EGFR activity or ISC mitoses³. The *SH3PX1* mutant effectively mimics activation of EGFR specifically in progenitors. As shown below, *SH3PX1*^{d1/10A} mutants specifically induced *egr* in progenitors, similar to targeted *Ras*^{V12S35} or *Raf*^{GOF} expression (Fig. 7a). Further, knockdown of *egr* in progenitors suppressed *SH3PX1*^{RNAi}-induced ISC hyperproliferation, confirming *egr* as a downstream target of SH3PX1. Hence, we used the *SH3PX1* mutant to mimic the overexpression of *egr* in progenitors. When we depleted *grnd* in ECs in these mutants with EC-targeted RNAi, this effectively blocked the ISC hyperproliferation in the *SH3PX1*^{d1/HK62b} mutants, affirming (albeit indirectly) that progenitor-derived Egr signals to Grnd in ECs. These data are included in the revision as new Supplementary Fig. 8.

- Line 165: wgn depletion in any cell type does not suppress ISC proliferation'. In fact, wgn knockdown in all cases shown has effects on ISCs that oppose those of *grnd* knockdown with more proliferation observed. In the former case. Why and is this Egr dependent?

Response: Thank you for bringing up this observation, which we also noted. To investigate this further, we did gain-of-function experiments, overexpressing *wgn* in progenitors. Interestingly,

overexpression of *wgn* in progenitors using *esg-Gal4^{ts}* did not show obvious pro-mitotic phenotypes, but it did markedly decrease damage-induced ISC proliferation. A recent study from the Colombani lab demonstrated that Grnd has a stronger binding affinity for Egr than Wgn⁴. Considering this, we hypothesize that Wgn competes with Grnd for Egr binding under stress conditions. Since Wgn's primary function in the fly midgut is not for ISC mitosis, but rather to regulate energy balance and immunity⁵, if Wgn binds more Egr, the pro-mitotic effect of Grnd/Egr should be reduced. This could explain why we observed more ISC proliferation in the *wgn^{RNAi}* group (Fig. 3f) but less ISC proliferation with *wgn* overexpression during stress conditions (Fig. 3k). Our new gain-of-function data on *wgn* is included in the revision as new Fig. 3k.

- The use of *Delta-gal4* as an ISC driver is problematic. This driver is well known for being patchy and, as stated by the authors themselves, weak when compared to the more robust and widely used *esg-gal4*, *Su(H)-gal80*, which is used in a few instances in the paper. Not sure why it has not been used consistently, instead of *Delta-gal4*. Related to this, the lack of phenotypes when using the *Delta-gal4* (e.g. following *grnd* overexpression or knockdown) are difficult to interpret and need to be re-assessed using *esg-gal4*, *Su(H)-gal80*.

Response: Thank you for highlighting this concern. While it is commonly understood in the field that *esg-gal4*, *Su(H)-gal80* is generally a stronger driver compared to *Dl-Gal4* for ISC-specific genetic manipulations, we have identified limitations with the *esg-Gal4*, *Su(H)-Gal80* driver due to its leaky expression in EBs in both the anterior and posterior midgut (as depicted in the figure below). Considering this issue, we still prefer to utilize the *Dl-Gal4* driver in our experiments, particularly for genes with distinct roles in ISCs versus EBs.

In addition, we used the *Dl-Gal4^{ts}* driver to repeat the experiment shown in the previous Supplementary Fig. S6a (now Fig. S7a), which originally utilized the *esg-Gal4^{ts}*, *Su(H)-Gal80* driver. We conducted a side-by-side comparison to evaluate the effects of both drivers when activating *UAS-rhomboid*. Interestingly, we did not observe a weaker effect with the *Dl-Gal4^{ts}* driver, attesting to its efficacy (see below).

- Lines 181-190: The rational and experimental design presented here is confusing as it is the main conclusion drawn from the results. Overexpression of *grnd* leading to JNK activation in ECs shows a non-cell autonomous and likely indirect effect of JNK activation by *grnd*, which does not relate to the intercellular signaling proposed in the study of cell autonomous activation of JNK by EC expressed *Grnd*.

Response: Thank you for bringing up this somewhat confusing issue. As we mentioned in our response to point #1, viewing this phenomenon from the perspective of a dynamic cell lineage with ongoing cell division and differentiation can provide clarity. Considering the transient nature of EBs, which rapidly differentiate into ECs, our interpretation is that overexpression of *grnd* using the *esg^{ts}* driver generates newborn ECs that inheriting the EB-overexpressed Grnd. Consequently, this activates the *puc-lacZ* reporter in such newborn ECs, as shown in Fig. 3o. In contrast, since ALG-mediated N-glycosylation appears to repress the interaction between Grnd and Egr in progenitors, *puc-lacZ* won't be induced when Grnd is overexpressed in these cells (indicated by yellow arrowheads in Fig. 3o). We have edited the manuscript to make this interpretation clearer (lines 184-194).

- Lines 204-205: Assessment of EC renewal following JNK activation in ISCs/EBs (Figure 4c-d). How was this experiment done? Lineage tracing is necessary for this type of analysis.

Response: For all quantifications of GFP intensity shown in Fig. 4d-e, the DAPI channel of each image was segmented using ImageJ. First, background fluorescence was removed via no-neighbor de-blurring, then images were thresholded, and a watershed algorithm was applied. After segmentation, the mean GFP fluorescence intensity was measured for all objects (*e.g.* nuclei) with minimum circularity of 0.5. To calculate the frequency of progenitors, marked by GFP expression, GFP fluorescence measurements from individual cells from the *w1118* control samples were thresholded using the Otsu's algorithm. Clearly visible cells with fluorescence levels above the threshold and a nuclear size $< 25\mu\text{m}^2$ were then marked as progenitors. We have provided detailed information about this procedure in the methods section of the paper (lines 535-543).

We agree that a lineage-tracing assay is a great approach here, and we tried several, one of which (G-TRACE) worked well and gave data that are now included in the paper. We utilized the G-TRACE system (genotype: *UAS-RFP, UAS-Flp, Ubi^{p63E}>FRT-STOP-FRT>nEGFP*)⁶ to track the cell fate of *Hep^{Act}*-expressing ISCs. G-TRACE lineage tracing was initiated by temperature shift using *esg^{ts}*. In this method, *UAS-RFP* and *UAS-Flp* are induced in progenitors by *esg-Gal4*. FLP recombinase excises the *FRT-STOP-FRT* cassette from *Ubi^{p63E}>FRT-STOP-FRT>nEGFP*, converting it to a stably expressed GFP marker, *Ubi^{p63E}-nEGFP*. As a result, all progenitors are marked with RFP while their progeny, descendent cells are marked with GFP only after differentiation. Under normal conditions, due to active gut regeneration, newborn ECs (nECs) will inherit RFP signals from their precursors, thereby exhibiting a mild-RFP/strong-GFP double positive pattern. In contrast, old pre-existing ECs (oECs) exhibit an RFP/GFP double negative pattern. When *Hep^{Act}* was overexpressed using the G-TRACE system, RFP⁺ progenitor cells rapidly disappeared and the number of nECs significantly decreased. This data confirms that hyperactivation of JNK in progenitors causes stem cell loss and impairs gut regeneration. This new data is included in the revision as new Fig. 4f.

- Figure 3n: Why is Grnd positioned in ISCs and not EBs where a function for the receptor has been shown in regeneration. This is also related to my first point.

Response: In fact, available data suggests that *grnd* is expressed in all midgut cell types ⁷, although our results indicate it is just not activated in ISCs. However, to avoid any confusion, we have removed this model panel from the revision. For additional clarification, please see our response to your first point.

- Figure 5: Role of Alg3 in the restriction of JNK activation.

a-*Alg3* is downregulated in ISCs/EBs following *Pe* or *Ecc15* infection. How do you reconcile this with the proposed model of JNK activation being inhibited by *Alg3* in ISCs/EBs given that JNK is not activated (especially in *Pe*) in these cells during infection in spite of *Alg3* being down?

Response: As shown in Fig. 5f, *Ecc15* infection induces a more pronounced reduction on *Alg3* and *Alg9* in progenitors as compared to *P.e.* infection. This suggests that the induction of *Alg3/9* by *P.e.* may not sufficiently impair the N-glycosylation of Grnd to activate the Egr/Grnd signaling in progenitors. To avoid any further confusion, we have removed the *P.e.* infection panel from the revised paper, as a similar phenotype has been reported in previous studies ^{8,9}. The remaining *Ecc15* infection data, now presented in new Fig. 2c, is adequate to support our main conclusion.

b-Also, if *Alg3* was the determinant factor as to whether JNK is activated or not downstream of Grnd, one would expect EC will be devoided of *Alg3*. Is this the case? What happens if *Alg3* is downregulated in ECs? Does this lead to JNK activation in homeostasis?

Response: These are very good questions. Currently, there are no antibodies available for ALG3 or ALG9, or established reporter lines to determine their expression patterns in the gut. However, our previous cell type-specific RNA-Seq data indicated that both *Alg3* and *Alg9* are expressed in all midgut cell types⁷. Regarding their functions in ECs, we have observed that depletion of *Alg3* or *Alg9* in ECs promotes ISC mitoses (Fig. 5e). Interestingly, inactivating JNK signaling in ECs with *Bsk^{K53R}* was insufficient to suppress *Alg9^{RNAi}*'s pro-mitotic effects (Fig. 5e), suggesting that ALG9 and/or ALG3 may have multiple glycosylation targets in ECs beyond Grnd, that affect ISC proliferation. It's also possible that the dysfunction in N-glycosylation (*e.g.* by *Alg3/9* depletion) in ECs induces EC stress, triggering a general damage response that promotes ISC proliferation. Once initiated, the hyper-proliferation of ISCs may stress their surrounding ECs (by displacing them), activating multiple stress signaling pathways such as Yki, P38, and JNK. Consequently, we anticipate that the JNK reporter *puc-lacZ* would be activated at a later stage of the response. However, the specific modification targets of ALG3/9 in ECs remain unknown, making it an intriguing topic for future study.

c- The data in Figure 5b shows that alg knockdown in progenitors also leads to predominant activation of JNK in EC. What does this mean? Overall, conclusions drawn from this data suffer from lack of experimental support.

Response: Similar to what we discussed in the above point #b, we believe the sequence of events probably unfolds as follows: Initially, *Alg3/9* knockdown in progenitors induces a cell-autonomous activation of JNK signaling, thereby triggering ISC proliferation. Subsequently, the overproliferated ISCs displace and stress their surrounding ECs, leading to the activation of multiple stress signaling pathways in ECs as a secondary response. Notably, JNK is among these stress signaling pathways. That's why *Alg3/9* knockdown in progenitors could result in the activation of *puc-lacZ* in ECs. We hope that all of our explanations above have helped the reviewer better understand our model. Thanks!

Minor comments: - Line 139: reference to previous work on haemocytes as a source Egr (30). The reference cited is not correct or at least not the first account of Egr in haemocytes, which came from Marcos Vidal's lab (Cordero et al, 2010-Dev Cell).

Response: Thanks for this correction. We have cited the Cordero/Vidal paper, replacing the previous one (line 137, new citation #31).

Reviewer #2:

In this manuscript, Zhang, Edgar and colleagues present a novel mechanism responsible for the increased proliferation of intestinal stem cells (ISCs) during aging and in the event of gut infection. They discovered that Egr (the *Drosophila* TNF), which is the ligand for the JNK signaling pathway, is specifically induced in intestinal progenitor cells as a result of aging or pathogenic infection. Egr then preferentially activates JNK signaling in the differentiated enterocytes by binding to the

Grnd receptor, leading to the upregulation of Rhomboid expression. This triggers the secretion of two growth factor receptor (GFR) ligands, Krn and Spi, which stimulate ISC proliferation through EGFR/Ras signaling. Interestingly, the study reveals that the expression of Egr in ISCs is promoted by EGFR/Ras signaling.

Based on their findings, the authors propose the existence of a positive feedback loop involving JNK-EGFR signaling between ISCs and enterocytes, which drives ISC proliferation in the aging or damaged gut. The authors also demonstrate that the N-glycosylation of Grnd, facilitated by Alg3 and Alg9 enzymes in ISCs, prevents autonomous JNK signaling activation within the ISCs. In the context of an infected gut, however, the downregulation of Alg3 enables JNK signaling activation in ISCs.

The manuscript is skillfully written, and overall, the mechanistic study is thorough and logically organized. Given the common occurrence of dysregulated TNF/TNFR and EGFR signaling in inflammatory diseases and cancer, the present findings hold significant implications for disease understanding and the advancement of potential treatments. Nevertheless, certain conclusions drawn from the data presented lack sufficient support and should be adequately addressed during the revision process.

Response: We appreciate the reviewer's positive assessment of our manuscript. We have now carefully revised this paper to address each of the reviewer's concerns. Please see the following point-by-point response.

Major points:

1. The authors propose that N-glycosylation of Grnd specifically occurs in ISCs. To confirm this, it is necessary to determine the expression pattern of Grnd and N-glycosylated Grnd in the intestinal epithelium.

Response: This is an important question. But unfortunately, we couldn't directly address it due to the lack of specific antibodies. However, we now provide some indirect evidence that is relevant to the reviewer's query. Previous research conducted in the fly wing discs by the David Bilder lab revealed that Grnd is N-glycosylated at its asparagine 63 by ALG3, which limits its binding to Egr. If this modification is conserved in the gut, wild-type Grnd should have a higher molecular weight than its mutant counterpart, Grnd^{N63A}, which lacks ALG-mediated glycosylation at N63. Our western blot data confirms this expectation: progenitor cell-expressed V5-tagged wild-type Grnd indeed displays a higher molecular weight than tagged Grnd^{N63A}, indicating that the N-glycosylation of Grnd at the N63 site normally occurs in progenitors (ISCs/EBs) in the fly midgut. We've included this new data in the revision as new Fig. 5j.

To test whether Grnd is glycosylated in ECs, we used the same type of test, but induced *UAS-grnd^{WT}-V5* or *UAS-grnd^{N63A}-V5* expression using the EC-specific *mex-Gal4^{ts}* driver. The result is similar to that from progenitor cells, although some minor extra higher molecular weight bands are seen for both forms of Grnd (see below, included in this file only). Thus, this new data supports our proposal that Grnd is glycosylated in the gut and that this restrains its activity, but it does not indicate that Grnd glycosylation is limited to ISCs. Accordingly, we have removed our proposal that ISC-specific Grnd glycosylation might account for the specific refractivity of ISCs to Egr→JNK signaling. In the revised manuscript, we highlight that this mechanism remains to be deciphered.

2. The authors highlight the significance of the expression level of *alg3* in regulating the specific pattern of N-glycosylated Grnd. It is important to investigate whether *alg3* is specifically expressed in ISCs and whether this pattern is disrupted in the infected gut. Also, is *alg3* overexpression able to inhibit JNK activation in ISCs of the infected gut?

Response: Another excellent question. We made considerable efforts to address this inquiry. However, due to the lack of an ALG3 antibody, we attempted to employ the RNA-fluorescence in situ hybridization (FISH) technique to detect *Alg3* expression in the gut with and without infection. Unfortunately, although many of our FISH probes work well in the midgut using a new HCR protocol, the FISH probe against *Alg3* did not yield satisfactory results. Nonetheless, our qPCR assay conducted in FACS-sorted progenitor cells revealed significant downregulation of *Alg3* by

infection (Fig. 5f). In addition, we repeated this qPCR assay during the revision, and our new data showed that not only *Alg3* but also *Alg9* was decreased by *Ecc15* and *P.e.* infection, indicating that both function as endogenous inhibitors for JNK activation in progenitor cells (see new Fig. 5f). Consistent with this, through checking our previous cell type-specific RNA-Seq data, we found that both *Alg3* and *Alg9* are significantly decreased in EBs after *P.e.* infection⁷. Furthermore, we attempted to use a western blot assay to investigate whether N-glycosylation of Grnd in progenitors is disrupted by *Alg3* or *Alg9* knockdown (strategy: *esg^{ts}*>*UAS-grnd^{WT}-v5*, *Alg3(or 9)^{RNAi}* versus *esg^{ts}*>*UAS-grnd^{N63A}-v5*, *Alg3(or 9)^{RNAi}* on molecular weights). Unfortunately, all transgenes, including *UAS-grnd*, *Alg3^{RNAi}*, and *Alg9^{RNAi}*, are targeted into the *atp40* landing site using the PhiC31 technique; we were unable to obtain the recombinants required for these experiments. We thank the reviewer for their understanding.

Also, as suggested by the reviewer, we conducted a gain-of-function assay of ALG3 in gut progenitors. Our data showed that overexpressing *Alg3* in progenitor cells significantly suppressed damage-induced ISC hyperproliferation (Fig. 5g, see data below). Together with other data in Fig. 5, this supports our proposal that ALG3 functions as an endogenous brake for JNK activation. We have included this new data in the revision as new Fig. 5g.

Further, we examined ALG3's gain-of-function role in ECs under both normal and stress conditions (see below, included in this file only). Unlike *grnd^{RNAi}*, overexpression of *Alg3* in ECs didn't repress damage-induced ISC proliferation. The mechanism behind this unexpected phenotype remains unclear, but it is consistent with our genetic epistasis test, which showed ALG3/9 may have multiple glycosylation targets in ECs beyond Grnd, that affect ISC proliferation (Fig. 5e, lines 265-266).

3. While Rhomboid is known to process EGFR ligand precursors, it is difficult to imagine to this reviewer why Rho overexpression in ECs has such a profound impact on ISC proliferation, especially considering the typically low expression of *spi* and *krn* in ECs in normal gut. Hence, it is worth investigating whether Rho overexpression in ECs causes transcriptional activation of *spi* and *krn*.

Response: Good point. To address this, we employed fluorescence in situ hybridization (HCR-FISH) to detect *spi* and *krn* mRNA in the gut with or without *Rho* overexpression. As shown below, at homeostasis *Krn* is predominantly expressed in ECs with some sporadic expression in progenitors (white arrowheads), while *spi* is expressed in both progenitors and ECs, with relatively higher levels in progenitors (yellow arrowheads). Upon *P.e.* infection, *spi* exhibited significant upregulation in progenitors, while *Krn* showed a mild upregulation in ECs. These expression patterns are consistent with previous reports, indicating that the probes against both *spi* and *Krn* were effective.

However, in guts overexpressing *Rho*, there were no significant changes in the expression levels of either *spi* or *krn*, suggesting that *Rho* overexpression in ECs does not induce transcriptional activation of *spi* and *krn*. Regarding the higher-than-expected mitosis levels observed in the *Rho* overexpressing guts, we attribute this phenomenon to the positive feedback effects we describe here and in previous papers. In this scenario, initially, overexpressing *Rho* in ECs induces the secretion of some Spi and Krn, which activates EGFR signaling in ISCs, thereby promoting ISC proliferation. Subsequently, the over-proliferation of ISCs exerts mechanical stress on the surrounding ECs (by displacing them), leading to the activation of multiple stress signals, including not only Egr in ISCs/EBs, but also Upd2/3 cytokines in ECs, as secondary responses that upregulate Spi, Krn and *Rho* expression. These secondary responses further amplify ISC proliferation, resulting in the observed higher levels of mitosis.

While we did not observe any changes in *spi* and *krn* upon Rho overexpression, previous RNA-Seq data has indicated that JNK activation significantly upregulates *krn* rather than *spi* (see below data). We believe this additional information may provide insight to the reviewer. Since the FISH data and RNA-Seq data are not directly relevant to the main narrative of the paper, we decided to include them only in this file and not in the main text. However we can include them in the paper if the review so desires.

4. Can JNK activation or rho overexpression in ECs trigger cell apoptosis, or is it dose- dependent?

Response: As shown in the following figure, the activation of JNK in ECs by *Hep^{Act}* overexpression triggers EC apoptosis, as indicated by DCP-1 staining. In contrast to progenitor cells, ECs exhibit a high sensitivity to JNK activation. A brief period (1-day) of *Hep^{Act}* expression in ECs induces robust EC apoptosis. However, over time (e.g., 3-day overexpression), there is no significant difference in the degree of EC apoptosis. This data has been included in the revised manuscript as the new Supplementary Fig. S4.

Minor points:

1. Figure 2, to demonstrate JNK activity also occurs in EEs, co-staining with an EE specific marker is necessary.

Response: Good point. We have addressed this by repeating the experiment and co-staining with Pros (an EE marker), Eiger-GFP, and Puc-lacZ. Please refer to the updated Figure, now presented as Fig. 2b in the revised manuscript.

2. Figure 4d, which cell type is positive for Dcp-1 staining, ISCs or ECs?

Response: Since *Hep^{Act}* is deleterious to progenitor cells, most of the *Hep^{Act}*-overexpressing progenitors lost their GFP labeling. However, upon careful examination of the nuclei size, we observed that DCP-1 staining was mainly evident in small and intermediate-sized cells, likely representing ISCs and EBs. Conversely, the largest cells (mature ECs) were mostly DCP-1 negative. In the revised Fig. 4g, we have added different colored arrowheads to indicate ISCs/EBs versus ECs.

3. Please describe the nature of the *egr* alleles: *egr^w* and *egr^{regg1}*

Response: *UAS-egr^w* refers to a standard UAS line in which the wildtype *egr* cDNA is cloned into the *pUAST* vector¹⁰. *UAS-egr^{regg1}* is a GS transposon insertion line (GS9830) that contains UAS enhancers inserted into the promoter region of *egr*¹⁰. This information has been included in the legend of Fig. 2.

4. Could the authors provide speculation on how the positive feedback loop is halted during the recovery from infection, allowing the event to cease?

Response: This is a very interesting question. We are currently conducting a genetic screen focused on the termination process of gut regeneration. So far, we have not determined which candidate identified from our screen could control this positive feedback loop. However, we speculate that the JNK downstream target, Puckered (Puc), which encodes Jun Kinase Phosphatase, may play a role in the termination of gut regeneration. Puc acts as a negative feedback inhibitor of the JNK pathway. We envision that once JNK is activated in ECs, Puc accumulates and suppresses JNK activity, leading to a decrease in the downstream target of JNK, Rho. Decreased Rho would then reduce secretion of Spi/Krn, which would in turn dampen EGFR signaling in progenitors. The downregulated EGFR signaling could then reduce Eiger production by progenitors, ultimately leading to the silencing of the Egr/Grnd/JNK-Rho-Krn/Spi-EGFR-Egr feedforward loop. We describe this possibility in the revised discussion section and hope this is a reasonable speculation. In principle, similar negative feedback in the Rho-EGFR pathway could squelch the positive feedback loop too, but the expression data we have give no indication that this happens.

Reviewer #3:

Drosophila intestinal stem cells (ISCs) represent a powerful model to elucidate the mechanisms controlling and executing regeneration fueled by somatic stem cells. This manuscript describes research that significantly extends a large body of already published information on the pathways that relay stress signals to the stem cells to initiate and control tissue repair.

It had been recognized for a while that JNK and Ras/Erk signaling are involved in the activation of ISCs in response to stress or tissue damage. The work described by Bruce Edgar and colleagues clarifies how the two signaling systems intersect. The JNK pathway is activated in the terminally differentiated enterocytes by the TNF protein Eiger secreted from adjacent ISCs or enteroblasts. This in turn causes the activation of the protease Rhomboid and the release of EGFR ligands back to the ISCs which triggers them to initiate proliferation and differentiation resulting in repair or tissue damage. This feed forward loop can be modulated by Alg3-mediated glycosylation which interferes with the productive interaction of Eiger with its receptor Grindelwald on ISCs.

The new information reported here is well supported by the experimental evidence. It represents a significant gain in the understanding of somatic stem cell-driven tissue repair and regeneration in the fly gut. The described paracrine feedback mechanism may operate similarly in other examples of stem cell function and, more generally, in diverse examples of physiological or pathological cell-cell interaction. General interest is high. Publication as a Nature communication paper is appropriate.

Response: We appreciate the reviewer's acknowledgement on the value of our experiments, and hope our revision will make our story even better.

Minor comment:

Line 238:

“...expression of BskK53R (a dominant-negative form Bsk/JNK) significantly blocked ISC mitoses caused by Alg9RNAi (Fig. 5d).” That is not shown in fig 5d. And the statement seems to be contradicted in the next sentence.

Response: Thanks for bringing this up. It should be Fig. 5e; we’ve corrected this mistake in the revised manuscript.

Reference

1. Kockel, L. *et al.* A Drosophila LexA Enhancer-Trap Resource for Developmental Biology and Neuroendocrine Research. *G3 (Bethesda)* **6**, 3017-3026 (2016).
2. Lee, J., Ng, K.G., Dombek, K.M., Eom, D.S. & Kwon, Y.V. Tumors overcome the action of the wasting factor ImpL2 by locally elevating Wnt/Wingless. *Proc Natl Acad Sci U S A* **118** (2021).
3. Zhang, P. *et al.* An SH3PX1-Dependent Endocytosis-Autophagy Network Restrains Intestinal Stem Cell Proliferation by Counteracting EGFR-ERK Signaling. *Dev Cell* **49**, 574-589 e575 (2019).
4. Palmerini, V. *et al.* Drosophila TNFRs Grindelwald and Wengen bind Eiger with different affinities and promote distinct cellular functions. *Nat Commun* **12**, 2070 (2021).
5. Loudhaief, R. *et al.* The Drosophila tumor necrosis factor receptor, Wengen, couples energy expenditure with gut immunity. *Sci Adv* **9**, eadd4977 (2023).
6. Evans, C.J. *et al.* G-TRACE: rapid Gal4-based cell lineage analysis in Drosophila. *Nat Methods* **6**, 603-605 (2009).
7. Dutta, D. *et al.* Regional Cell-Specific Transcriptome Mapping Reveals Regulatory Complexity in the Adult Drosophila Midgut. *Cell Rep* **12**, 346-358 (2015).
8. Apidianakis, Y., Pitsouli, C., Perrimon, N. & Rahme, L. Synergy between bacterial infection and genetic predisposition in intestinal dysplasia. *Proc Natl Acad Sci U S A* **106**, 20883-20888 (2009).
9. Jiang, H. *et al.* Cytokine/Jak/Stat signaling mediates regeneration and homeostasis in the Drosophila midgut. *Cell* **137**, 1343-1355 (2009).
10. Igaki, T. *et al.* Eiger, a TNF superfamily ligand that triggers the Drosophila JNK pathway. *EMBO J* **21**, 3009-3018 (2002).

Reviewer #1 (Remarks to the Author):

The revised version of this manuscript addresses my major comments with either experiments or text clarifications. I am therefore satisfied with this version and have no additional requests..

Reviewer #2 (Remarks to the Author):

The authors have adequately addressed all the concerns raised by the reviewers and have made modifications to some conclusions based on the new data. The findings presented in the paper are both interesting and important, and I believe the paper is now ready for publication